# Thermodynamic theory of inverse current in coupled quantum transport

Shuvadip Ghosh,[1] Nikhil Gupt,[1] and Arnab Ghosh[1, *]

[1]*Indian Institute of Technology Kanpur, Kanpur, Uttar Pradesh 208016, India*

The inverse current in coupled (ICC) quantum transport, where one induced current opposes all thermodynamic forces of a system, is a highly counter-intuitive transport phenomenon. Using an exactly solvable model of strongly-coupled quantum dots, we present thermodynamic description of ICC in energy and spin-induced particle currents, with potential applications towards unconventional and autonomous nanoscale thermoelectric generators. Our analysis reveals the connection between microscopic and macroscopic formulations of entropy production rates, elucidating the often-overlooked role of proper thermodynamic forces and conjugate fluxes in characterizing genuine ICC. In our model, the seemingly paradoxical results of ICC in the energy current arise from chemical work done by current-carrying quantum particles, while in spin-induced particle current, it stems from the relative competition between electron reservoirs controlling one particular transition.

## I. INTRODUCTION

Force and flux are two central thermodynamic quantities interlinked through a cause-and-effect relationship. Thermodynamic force emerges due to a gradient in the system and engenders a response in the form of flux within the system, causing a departure from equilibrium. Consequently, the resultant flux aligns in the direction of the applied force to establish a new equilibrium. One of the most intriguing transport phenomena is absolute negative mobility (ANM), wherein the system's response i.e., the generated current, operates against the driving force [1–3]. The absence of ANM, at near-equilibrium for a single flux-force pair, can be understood from the sign of the entropy production rate ($\dot{\Sigma} = J\mathcal{F}$) to remain consistent with the second law, where $J$ represents the thermodynamic flux conjugate to the thermodynamic force $\mathcal{F}$. Initially, ANM was considered as a consequence of the quantum effect [4, 5]. However, theoretical works revealing the lack of any fundamental laws prohibiting ANM in systems far from equilibrium created significant interests [1–3, 6–10], which have been verified in experiments [5, 11, 12].

The above situation changes dramatically in the presence of coupled transports with multiple force-flux pairs, where the entropy production rate near equilibrium takes the general form $\dot{\Sigma} = \sum_i J_i\mathcal{F}_i$. This could in principle lead to the concept of Inverse Current in Coupled transport (ICC), where a current flows against all forces, including its own conjugate force. It is crucial to distinguish ICC from a *cross-effect*, where a current is produced in response to a non-conjugate force, despite the fact that its own conjugate force is set to zero. While a cross-effect (Seebeck and Peltier) is a regular outcome in coupled transport, ICC involves a current opposing to all forces present in the system, including its own conjugate force, appearing highly counterintuitive, yet not forbidden as long as the overall entropy production rate remains in the positive domain. To examine ICC, one must therefore consider at least two nonzero forces and their corresponding fluxes. For example, the thermal (energy) force $\mathcal{F}_E$, resulting from a temperature gradient ($\Delta T$) and the particle force $\mathcal{F}_N$, arising from chemical potential gradient ($\Delta\mu$), could drive both particle ($J_N$) and energy ($J_E$) currents. As a result, entropy production rate is expressed as $\dot{\Sigma} = J_E\mathcal{F}_E + J_N\mathcal{F}_N$. When both currents may counter-operate their non-conjugate forces, for ICC to occur, either $J_E$ or $J_N$ must exhibit a sign opposite to both forces. The immediate consequence is that simultaneous ICC in both fluxes is impossible, as it would violate the second law of thermodynamics, resulting in a negative entropy production rate.

Recently, based on a classical Lieb-Liniger model [13], characterized by 1D-interacting Hamiltonian of a diatomic gas, a potential platform for ICC, separately for energy and particle currents, has been demonstrated by Wang *et al.* [14]. The seemingly paradoxical result in the above classical model originates from the occurrence of self-organization within the system in response to applied forces. Since, the general criteria for ICC are yet unknown either in classical/quantum settings, showcasing the existence of ICC in quantum systems, still remains a difficult task. Only a few attempts are made [15, 16] which however fell short of establishing ICC in terms of proper thermodynamic forces. Though numerical results managed to produce the flow of one current against both gradients ($\Delta T$ and $\Delta\mu$), these observations do not qualify as ICC, as thermodynamic forces and gradients are altogether distinct quantities. Thus, realizing genuine ICC in quantum systems with appropriate thermodynamic analysis still remains a highly challenging endeavor.

In this paper, we provide a resolution to this problem by constructing a thermodynamic theory of inverse current using an exactly solvable model of a three-terminal Coulomb-coupled quantum dots (QDs), a simple variant of the extensively studied Sánchez-Büttiker model [17], explored in number of contexts, such as quantum transport [18–24], quantum information [25–27] and thermoelectricity [28–41], thermal rectification [42–48], and many others [49–51]. Our findings unravel the crucial role played by the QD interaction in establishing ICC in en-

* arnab@iitk.ac.in

ergy and particle currents. The key element of our analysis lies in identifying macroscopic and microscopic formulas for thermodynamic forces and fluxes. Most importantly, we have established conditions for genuine ICC in both energy and (spin-polarized) particle currents, without violating the second law of thermodynamics.

The structure of the paper is as follows: Section II outlines our model, which is based on Coulomb-coupled QDs. We explore the system's dynamics in Sec. III, analyze steady-state currents in Sec. IV, and examine entropy production, including macroscopic and microscopic forces and fluxes, in Sec. V. Section VI provides a summary of our results. Finally, we conclude in Sec. VII.

## II. THEORETICAL MODEL

Our model consists of two strongly and capacitively coupled quantum dots (QDs), labeled by left (QD$_L$) and right (QD$_R$), respectively [FIG. 1]. These dots solely interact via a long-range Coulomb force restricting particle exchange due to Coulomb blockade [42, 45, 48] while facilitating energy exchange within the dots through Coulomb interaction ($\kappa_c$). The QD$_L$ is simultaneously tunnel-coupled with two fermionic reservoirs [17, 48, 52, 53], labeled above ($a$) and below ($b$), allowing particle flow between the two reservoirs. The QD$_R$ is tunnel-coupled with only one reservoir ($r$), permitting both particle and energy exchange between the dot and the coupled lead. While for spinless electrons, $\kappa_c$ is always positive, w.l.o.g, we consider spin-polarized [54–56] electrons to account for a more comprehensive study. To facilitate electron exchange between the leads $a$ and $b$ through QD$_L$, we assume that reservoirs $a$ and $b$ consist of only spin-down ($\downarrow$) electrons, while reservoir $r$ is filled with spin-up ($\uparrow$) electrons. The setup could lead to both attractive (negative) and repulsive (positive) interaction between the QDs depending on the relative strength of the $\kappa_c$ and the spin-spin interaction ($\kappa_s$). The Hamiltonian governing the coupled QD system is given by [43, 48, 57, 58],

$$H_s = \varepsilon_L \mathcal{N}_{L\downarrow} + \varepsilon_R \mathcal{N}_{R\uparrow} + \kappa_c \mathcal{N}_{L\downarrow} \mathcal{N}_{R\uparrow} + \kappa_s \sigma_{L\downarrow}^z \sigma_{R\uparrow}^z. \quad (1)$$

In the above Hamiltonian, $\varepsilon_\alpha$ ($\alpha = L, R$), represents the single-particle energy level associated with the $\alpha$'th QD. Due to the Coulomb blockade, the electron density in the dots is low, restricting occupancy to either zero or one. Under this condition, the eigenstates of QD$_\alpha$ are either $|0\rangle$ or $|\downarrow\rangle$ ($|\uparrow\rangle$), with energy eigenvalues 0 and $\varepsilon_L$($\varepsilon_R$), respectively. The corresponding number operators for QD$_\alpha$ are $\mathcal{N}_{L\downarrow} = d_{L\downarrow}^\dagger d_{L\downarrow}$ and $\mathcal{N}_{R\uparrow} = d_{R\uparrow}^\dagger d_{R\uparrow}$, where $d_{L\downarrow}^\dagger$ ($d_{L\downarrow}$) and $d_{R\uparrow}^\dagger$ ($d_{R\uparrow}$) denote the electron creation (annihilation) operators for the respective QD$_\alpha$, obeying to the anticommutation relations $\{d_{L\downarrow}, d_{L\downarrow}^\dagger\} = 1 = \{d_{R\uparrow}, d_{R\uparrow}^\dagger\}$. The last term in Eq. (1) represents the spin-spin interaction energy where $\sigma_{L\downarrow(R\uparrow)}^z = 1 \mp 2\mathcal{N}_{L\downarrow(R\uparrow)}$, satisfying standard operator algebra $\sigma_{L\downarrow}^z|\downarrow\rangle = -1|\downarrow\rangle$ and $\sigma_{R\uparrow}^z|\downarrow\rangle = +1|\uparrow\rangle$.

As the two QDs are strongly and capacitively coupled, the overall system Hamiltonian is diagonal in the eigenbasis of the individual QD. This can be represented by the tensor product of the number operator's eigenbasis of the coupled QDs. For convenience, the four eigenstates $\{|0\rangle, |\downarrow\rangle\} \otimes \{|0\rangle, |\uparrow\rangle\}$, are labeled by $|\mathbb{A}\rangle = |00\rangle$, $|\mathbb{B}\rangle = |\downarrow 0\rangle$, $|\mathbb{C}\rangle = |0\uparrow\rangle$, $|\mathbb{D}\rangle = |\downarrow\uparrow\rangle$ and their corresponding eigenenergies ($\varepsilon_{\mathbb{i}}, \mathbb{i} = \mathbb{A}, \mathbb{B}, \mathbb{C}, \mathbb{D}$) are $\varepsilon_{\mathbb{A}} = 0$, $\varepsilon_{\mathbb{B}} = \varepsilon_L$, $\varepsilon_{\mathbb{C}} = \varepsilon_R$ and $\varepsilon_{\mathbb{D}} = \varepsilon_L + \varepsilon_R + \kappa$ respectively [FIG. 1: Inset]. Consequently, the energy of the most excited state of the composite system is $\varepsilon_L + \varepsilon_R + \kappa$, where, we assume w.l.o.g, $\varepsilon_L < \varepsilon_R$. Although, $\kappa_c$ and $\kappa_s$ are non-negative quantities, the overall interaction $\kappa = \kappa_c - \kappa_s$, can take any real values involving positive and negative domains of interaction. It is important to note that when $\kappa$ is negative and $|\kappa| > \varepsilon_L$, the energy states $|\mathbb{C}\rangle$ and $|\mathbb{D}\rangle$ swap their positions [FIG. 1: Inset]. This will play a crucial role in exhibiting ICC behavior, as explored in Sec. VI.

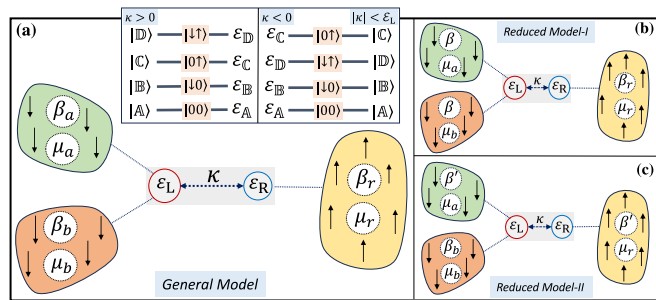

FIG. 1. Main: **(a)** Schematic diagram of the three-terminal Coulomb-coupled QDs. Inset: The energy level diagram of the eigenstates of coupled the QDs for $\kappa > 0$ (left) and $\kappa < 0; |\kappa| > \varepsilon_L$ (right). From the general model **(a)**, we can construct two reduced models:- **(b)** Reduced model-I, by considering the temperatures of the reservoirs ($a$) and ($b$) equal. **(c)** Reduced model-II, by setting up the temperatures of the reservoirs ($a$) and ($r$) the same.

Finally, the reservoirs are dense with fermionic particles, i.e., electrons, characterized by temperature and chemical potential. The Hamiltonian of the $\lambda$'th spin-polarized reservoir with electron spin $\sigma = \{\uparrow, \downarrow\}$, is defined as $H_B^{\lambda\sigma} = \sum_k (\epsilon_k^{\lambda\sigma} - \mu_{\lambda\sigma}) c_{\lambda\sigma k}^\dagger c_{\lambda\sigma k}$, where $\epsilon_k^\lambda$ is the energy of the non-interacting electrons for the reservoir $\lambda$, $k$ is the continuous wavenumber, $\mu_{\lambda\sigma}$ is the chemical potential, and $c_{\lambda\sigma}^\dagger (c_{\lambda\sigma})$ represents the creation (annihilation) operator. The total Hamiltonian of all three reservoirs is given by $H_B = H_B^{a\downarrow} + H_B^{b\downarrow} + H_B^{r\uparrow}$. The QD$_R$ (QD$_L$) is weakly coupled to the reservoir(s) $r$ ($a$ and $b$) to ensure sequential tunneling [45, 48, 53], such that only one QD at a time is involved with particle tunneling with the coupled lead. The tunnel-coupled Hamiltonians are characterized by the coupling constant $t_k^{\alpha\sigma\lambda}$ and are given

by

$$
\begin{aligned}
H_{\mathrm{T}}^{\mathrm{L}\downarrow a(b)} &= \hbar \sum_k [t_k^{\mathrm{L}\downarrow a(b)} c_{a(b)\downarrow k}^\dagger d_{\mathrm{L}\downarrow} + t_k^{\mathrm{L}\downarrow a(b)*} d_{\mathrm{L}\downarrow}^\dagger c_{a(b)\downarrow k}], \\
H_{\mathrm{T}}^{\mathrm{R}\uparrow r} &= \hbar \sum_k [t_k^{\mathrm{R}\uparrow r} c_{r\uparrow k}^\dagger d_{\mathrm{R}\uparrow} + t_k^{\mathrm{R}\uparrow r*} d_{\mathrm{R}\uparrow}^\dagger c_{r\uparrow k}].
\end{aligned} \quad (2)
$$

where, $H_{\mathrm{T}}^{\mathrm{L}\downarrow a(b)}$ signifies the interaction between QD$_{\mathrm{L}}$ and reservoir $a(b)$ with the exchange of $\downarrow$ electrons and $H_{\mathrm{T}}^{\mathrm{R}\uparrow r}$ represents the interaction between QD$_{\mathrm{R}}$ and reservoir $r$ through the exchange of $\uparrow$ electrons. Further, the transitions between $|\mathbb{B}\rangle \leftrightarrow |\mathbb{C}\rangle$ and $|\mathbb{A}\rangle \leftrightarrow |\mathbb{D}\rangle$ are prohibited due to Coulomb blockade and the sequential tunneling approximation [45, 48, 53], respectively. So, there are in total four authorized transitions between the eigenstates of the composite system, and each of the three reservoirs $a$, $b$, and $r$ controls two transitions, i.e., $|\mathbb{A}\rangle \leftrightarrow |\mathbb{B}\rangle$ and $|\mathbb{C}\rangle \leftrightarrow |\mathbb{D}\rangle$ are triggered by leads $a$ and/or $b$, while reservoir $r$ governs the transitions between $|\mathbb{A}\rangle \leftrightarrow |\mathbb{C}\rangle$ and $|\mathbb{B}\rangle \leftrightarrow |\mathbb{D}\rangle$. The transition energy between the eigenstates $|\mathbb{i}\rangle \rightarrow |\mathbb{j}\rangle$ is defined as $\omega_{\mathbb{ij}} = \varepsilon_{\mathbb{j}} - \varepsilon_{\mathbb{i}}$, where $\varepsilon_{\mathbb{i}}$ is the eigenenergy of $H_{\mathrm{s}}$ with the eigenstate $\{|\mathbb{i}\rangle\}$ and the four transition energies are respectively $\omega_{\mathbb{AB}} = \varepsilon_{\mathrm{L}}$, $\omega_{\mathbb{AC}} = \varepsilon_{\mathrm{R}}$, $\omega_{\mathbb{CD}} = \varepsilon_{\mathrm{L}} + \kappa$, and $\omega_{\mathbb{BD}} = \varepsilon_{\mathrm{R}} + \kappa$. To investigate the system's dynamics, we will next evaluate the transition rates using the Lindblad Master Equation (LME) under the Born-Markov approximation.

## III. EVALUATION OF RATE EQUATIONS USING LME

The state of the composite system is described by the reduced density matrix $\rho_{\mathrm{s}}(t) = \mathrm{Tr}_{\mathrm{B}}\{\rho_{\mathrm{tot}}(t)\}$, where $\rho_{\mathrm{tot}}(t)$ is the total density matrix of the system and bath combined. We utilize strong-coupling formalism to derive the LME governing the time evolution of the reduced density matrix $\rho_{\mathrm{s}}(t)$ of the system, under the Born, Markov, and Secular (BMS) approximation [48, 57, 59, 60] (Appendix A):

$$
\frac{d}{dt}\rho_{\mathrm{s}}(t) = \sum_\lambda \mathcal{L}_\lambda[\rho_{\mathrm{s}}(t)]; \quad \lambda = a, b, r. \quad (3)
$$

It is worth noting that the strong coupling formalism pertains to the interaction between the QDs, while we maintain the assumption of weak coupling between the system and its surrounding environment. This allows us to safely implement the BMS approximation and derive the above LME [Eq. (3)] purely based on the eigenstates of the full system Hamiltonian $H_{\mathrm{s}}$. As a result, the dissipation mechanism of each QD is influenced not only by its coupling to its individual bath but also by the interactions between QDs themselves. This consideration appears crucial for accurately characterizing heat flow and obtaining outcomes across a wide range of system parameters, as examined below. $\mathcal{L}_\lambda[\rho_{\mathrm{s}}(t)]$ in the above equation is the Lindblad super-operator, defined as

$$
\begin{aligned}
\mathcal{L}_\lambda[\rho_{\mathrm{s}}(t)] = \sum_{\{\omega_\alpha\}>0} &\left\{ \gamma_\lambda(\omega_\alpha) f_\lambda^+(\omega_\alpha) \left[ d_{\alpha\sigma}^\dagger(\omega_\alpha) \rho_{\mathrm{s}} d_{\alpha\sigma}(\omega_\alpha) - \frac{1}{2}\{\rho_{\mathrm{s}}, d_{\alpha\sigma}(\omega_\alpha) d_{\alpha\sigma}^\dagger(\omega_\alpha)\} \right] \right. \\
&\left. + \gamma_\lambda(\omega_\alpha) f_\lambda^-(\omega_\alpha) \left[ d_{\alpha\sigma}(\omega_\alpha) \rho_{\mathrm{s}} d_{\alpha\sigma}^\dagger(\omega_\alpha) - \frac{1}{2}\{\rho_{\mathrm{s}}, d_{\alpha\sigma}^\dagger(\omega_\alpha) d_{\alpha\sigma}(\omega_\alpha)\} \right] \right\},
\end{aligned} \quad (4)
$$

where $\gamma_\lambda(\omega_\alpha)$ is the bare electron transfer rate between the reservoir $\lambda$ and coupled QD$_\alpha$. The explicit forms in terms of the system-reservoir coupling constants can be calculated using Fermi's golden rule, as $\gamma_\lambda(\omega_\alpha) \equiv \gamma_{\lambda\sigma}(\omega_\alpha) = 2\pi \sum_k |t_k^{\alpha\sigma\lambda}|^2 \delta(\omega_\alpha - \epsilon_k^{\lambda\sigma})$, where $\omega_\alpha$ represents the required amount of energy associated with the transition between QD$_\alpha$ and its coupled lead. For simplicity of notations, we use $\gamma_{a|b}(\omega_{\mathrm{L}}) \equiv \gamma_{a|b}$ and $\gamma_r(\omega_{\mathrm{R}}) \equiv \gamma_r$, where $\omega_{\mathrm{L}} = \{\omega_{\mathbb{AB}}, \omega_{\mathbb{CD}}\}$ and $\omega_{\mathrm{R}} = \{\omega_{\mathbb{AC}}, \omega_{\mathbb{BD}}\}$, if the particle is entering (excitation) into the system, and similarly, for particle de-excitation (leaving) w.r.t. system, $\omega_{\mathrm{L}} = \{\omega_{\mathbb{BA}}, \omega_{\mathbb{DC}}\}$ and $\omega_{\mathrm{R}} = \{\omega_{\mathbb{CA}}, \omega_{\mathbb{DB}}\}$. Here, the symbol $a|b$ denotes the transition controlled by either $a$ or $b$ bath. Finally, $f_\lambda^\pm(\omega_{\mathbb{ij}})$ represents the Fermi distribution function (FDF) related to the transition $|\mathbb{i}\rangle \rightarrow |\mathbb{j}\rangle$ driven by the $\lambda$'th reservoir which cost $\omega_{\mathbb{ij}}$ amount of energy. The notation '$\pm$' which is used throughout the text [15, 17], where the '$\pm$' sign refers to particle excita-

tion (entering into the system) and the '$-$' sign refers to particle de-excitation (leaving from the system) w.r.t the system. Hence, if the transition from $|\mathbb{i}\rangle$ to $|\mathbb{j}\rangle$ is governed by the particle excitation w.r.t the system then the reverse process would signify the tunneling of the particle from the system to the environment. Thus, we establish the following relation between the FDFs associated with the above transitions [47, 48],

$$
f_\lambda^+(\omega_{\mathbb{ij}}) = 1 - f_\lambda^-(\omega_{\mathbb{ji}}) = \left[ 1 + \exp\left( \frac{\omega_{\mathbb{ij}} - \mu_\lambda}{k_B T_\lambda} \right) \right]^{-1}, \quad (5)
$$

where $T_\lambda$ and $\mu_\lambda$ are respectively the temperature and chemical potential of the fermionic reservoir $\lambda$. For the sake of convenience, in Eq. (5), we have dropped the $\sigma$ from the temperature and chemical potential. Anticipating what will become clear as we proceed, the above notation has the advantage that it does not rely on the specific energy level ordering of the system eigenstates. There-

fore, all subsequent calculations can be done in a compact form without referring to any particular arrangement of the system eigen-energies.

So, the time evolution of the occupation probabilities which are the diagonal elements of the reduced density matrix, i.e., $\rho_{\mathbb{i}\mathbb{i}} = \langle \mathbb{i}|\rho_{\mathrm{s}}(t)|\mathbb{i}\rangle$, can be evaluated from the LME [Eq. (3)] as

$$\dot{\rho}_{\mathbb{i}} \equiv \frac{d\rho_{\mathbb{i}\mathbb{i}}}{dt} = \langle \mathbb{i}|\dot{\rho}_{\mathrm{s}}(t)|\mathbb{i}\rangle = \sum_{\lambda=a,b,r} \langle \mathbb{i}|\mathcal{L}_{\lambda}[\rho_{\mathrm{s}}(t)]|\mathbb{i}\rangle. \quad (6)$$

Using the Lindbladians defined in Eq. (4), we can determine the rate equations as follows

$$\begin{aligned}
\dot{\rho}_{\mathbb{A}} &= -\Gamma_{\mathbb{AB}}^{a+} - \Gamma_{\mathbb{AB}}^{b+} - \Gamma_{\mathbb{AC}}^{r+} \equiv -\Gamma_{\mathbb{AB}}^{ab+} - \Gamma_{\mathbb{AC}}^{r+}; \\
\dot{\rho}_{\mathbb{B}} &= \Gamma_{\mathbb{AB}}^{a+} + \Gamma_{\mathbb{AB}}^{b+} - \Gamma_{\mathbb{BD}}^{r+} \equiv \Gamma_{\mathbb{AB}}^{ab+} - \Gamma_{\mathbb{BD}}^{r+}; \\
\dot{\rho}_{\mathbb{C}} &= -\Gamma_{\mathbb{CD}}^{a+} - \Gamma_{\mathbb{CD}}^{b+} + \Gamma_{\mathbb{AC}}^{r+} \equiv -\Gamma_{\mathbb{CD}}^{ab+} + \Gamma_{\mathbb{AC}}^{r+}; \\
\dot{\rho}_{\mathbb{D}} &= \Gamma_{\mathbb{CD}}^{a+} + \Gamma_{\mathbb{CD}}^{b+} + \Gamma_{\mathbb{BD}}^{r+} \equiv \Gamma_{\mathbb{CD}}^{ab+} + \Gamma_{\mathbb{BD}}^{r+},
\end{aligned} \quad (7)$$

where, $\Gamma_{\mathbb{i}\mathbb{j}}^{ab\pm} \equiv \Gamma_{\mathbb{i}\mathbb{j}}^{a\pm} + \Gamma_{\mathbb{i}\mathbb{j}}^{b\pm}$. The net transition rate $\Gamma_{\mathbb{i}\mathbb{j}}^{\lambda\pm}$ from level $|\mathbb{i}\rangle$ to $|\mathbb{j}\rangle$ mediated by the reservoir $\lambda$ ($\lambda = a, b, r$) for both the '$\pm$' processes can be expressed in a compact form as

$$\begin{aligned}
\Gamma_{\mathbb{i}\mathbb{j}}^{\lambda\pm} \equiv \Gamma_{\mathbb{i}\mapsto\mathbb{j}}^{\lambda\pm} &= \gamma_{\lambda} f_{\lambda}^{\pm}(\omega_{\mathbb{i}\mathbb{j}})\rho_{\mathbb{i}} - \gamma_{\lambda} f_{\lambda}^{\mp}(\omega_{\mathbb{j}\mathbb{i}})\rho_{\mathbb{j}}, \\
&= \mathrm{k}_{\mathbb{i}\mathbb{j}}^{\lambda\pm}\rho_{\mathbb{i}} - \mathrm{k}_{\mathbb{j}\mathbb{i}}^{\lambda\mp}\rho_{\mathbb{j}}
\end{aligned} \quad (8)$$

where $\Gamma_{\mathbb{i}\mathbb{j}}^{\lambda+} = -\Gamma_{\mathbb{j}\mathbb{i}}^{\lambda-}$ and

$$\mathrm{k}_{\mathbb{i}\mathbb{j}}^{\lambda\pm} = \gamma_{\lambda} f_{\lambda}^{\pm}(\omega_{\mathbb{i}\mathbb{j}}) \quad ; \quad \mathrm{k}_{\mathbb{j}\mathbb{i}}^{\lambda\mp} = \gamma_{\lambda} f_{\lambda}^{\mp}(\omega_{\mathbb{j}\mathbb{i}}). \quad (9)$$

Using Eq. (5), one can verify that the above rate coefficients for fermionic reservoirs satisfy an interesting rela-

tion

$$\mathrm{k}_{\mathbb{i}\mathbb{j}}^{\lambda\pm} + \mathrm{k}_{\mathbb{j}\mathbb{i}}^{\lambda\mp} = \gamma_{\lambda}. \quad (10)$$

For simplicity, we further assume all $\gamma_{\lambda}$ are equal when calculating the closed-form analytical expression for the steady-state transition rates in Sec. IV.

## IV. STEADY STATE CURRENTS AND DYNAMICS

In this section, we derive the expressions for steady-state currents under the grand canonical formalism, since the composite system weakly interacts with multiple reservoirs and exchanges both the energy and particles. The initial density operator of the system $\rho_{\mathrm{s}}(0)$ at equilibrium can be expressed as [60]

$$\rho_{\mathrm{s}}^{\mathrm{eq}} = \frac{e^{-\beta(H_{\mathrm{s}} - \mu\mathcal{N})}}{\mathcal{Z}(\beta, \mu)}, \quad (11)$$

where $\mathcal{Z}(\beta, \mu) = \mathrm{Tr}\left[e^{-\beta(H_{\mathrm{s}} - \mu\mathcal{N})}\right]$ is the grand canonical partition function and $\mathcal{N} = \mathcal{N}_{\mathrm{L}\downarrow} + \mathcal{N}_{\mathrm{R}\uparrow}$ being the total particle number operator of the two QDs. The $\beta$ and $\mu$ are the effective inverse temperature and chemical potential, $\beta = \sum_{\lambda}\beta_{\lambda}$ and $\mu = \sum_{\lambda}\mu_{\lambda}$ respectively. We assume that environmental interaction slightly perturbs the system from its initial equilibrium state $\rho_{\mathrm{s}}(0) = \rho_{\mathrm{s}}^{\mathrm{eq}}$ to $\rho_{\mathrm{s}}(t)$, such that $\delta\rho_{\mathrm{s}}(t) = \rho_{\mathrm{s}}(t) - \rho_{\mathrm{s}}(0) \equiv \mathcal{O}(\xi)$, where $\xi$ is a small expansion parameter. Equating system von-Neumann entropy (times $k_B$) with thermodynamic entropy near close to equilibrium [61–63]

$$\mathcal{S}_{\mathrm{s}}(t) = -k_B \mathrm{Tr}_{\mathrm{s}}[\rho_{\mathrm{s}}(t) \ln \rho_{\mathrm{s}}(t)], \quad (12)$$

one can derive the following expression for $\Delta\mathcal{S}_{\mathrm{s}}(t) = \mathcal{S}_{\mathrm{s}}(t) - \mathcal{S}_{\mathrm{s}}(0)$, considering up to first order change in $\xi$,

$$\begin{aligned}
\Delta\mathcal{S}_{\mathrm{s}}(t) =& -k_B \mathrm{Tr}_{\mathrm{s}}[\delta\rho_{\mathrm{s}}(t) \ln \rho_{\mathrm{s}}(0)] - k_B \mathrm{Tr}_{\mathrm{s}}\left[\delta\rho_{\mathrm{s}}(t) \ln\left\{1 + \frac{\delta\rho_{\mathrm{s}}(t)}{\rho_{\mathrm{s}}(0)}\right\}\right] - k_B \mathrm{Tr}_{\mathrm{s}}\left[\rho_{\mathrm{s}}(0) \ln\left\{1 + \frac{\delta\rho_{\mathrm{s}}(t)}{\rho_{\mathrm{s}}(0)}\right\}\right] \\
=& -k_B \mathrm{Tr}_{\mathrm{s}}[\delta\rho_{\mathrm{s}}(t) \ln \rho_{\mathrm{s}}^{\mathrm{eq}}] + \mathcal{O}(\xi^2) \\
\equiv& \, k_B\beta \mathrm{Tr}_{\mathrm{s}}[\delta\rho_{\mathrm{s}}(t)H_{\mathrm{s}}] - k_B\beta\mu \mathrm{Tr}_{\mathrm{s}}[\delta\rho_{\mathrm{s}}(t)\mathcal{N}].
\end{aligned} \quad (13)$$

Comparing the above equation and the first law of thermodynamics in the presence of *chemical work done*, we identify the change in energy and particle number as follows

$$\Delta E = \mathrm{Tr}_{\mathrm{s}}[\delta\rho_{\mathrm{s}}(t)H_{\mathrm{s}}] \quad ; \quad \Delta N = \mathrm{Tr}_{\mathrm{s}}[\delta\rho_{\mathrm{s}}(t)\mathcal{N}]. \quad (14)$$

Noting $\delta\rho_{\mathrm{s}}(t) = \rho_{\mathrm{s}}(t) - \rho_{\mathrm{s}}(0)$, and $H_{\mathrm{s}}$, $\mathcal{N}$ are time-independent, we immediately recognize the net energy

flux ($J_{\mathrm{E}}$) and particle flux ($J_{\mathrm{N}}$) as

$$\begin{aligned}
J_{\mathrm{E}}(t) &= \mathrm{Tr}_{\mathrm{s}}[\dot{\rho}_{\mathrm{s}}(t)H_{\mathrm{s}}] = \sum_{\lambda=a,b,r} \mathrm{Tr}_{\mathrm{s}}[\mathcal{L}_{\lambda}[\rho_{\mathrm{s}}(t)]H_{\mathrm{s}}], \\
J_{\mathrm{N}}(t) &= \mathrm{Tr}_{\mathrm{s}}[\dot{\rho}_{\mathrm{s}}(t)\mathcal{N}] = \sum_{\lambda=a,b,r} \mathrm{Tr}_{\mathrm{s}}[\mathcal{L}_{\lambda}[\rho_{\mathrm{s}}(t)]\mathcal{N}],
\end{aligned} \quad (15)$$

where we use Eq. (3) for $\dot{\rho}_{\mathrm{s}}(t)$. Identifying $J_{\mathrm{E(N)}}$ as the sum of the contributions of energy (particle) flux associ-

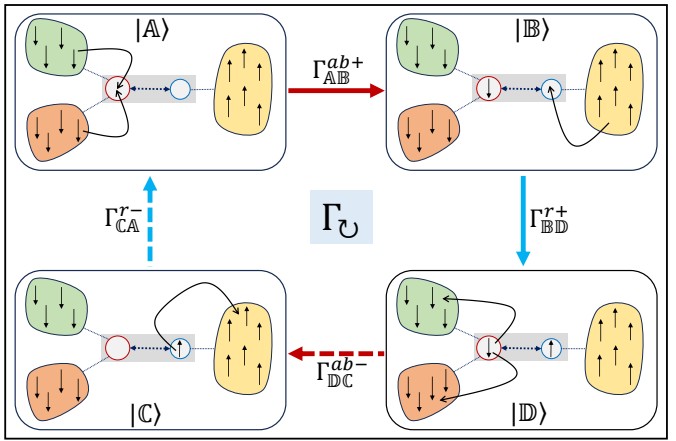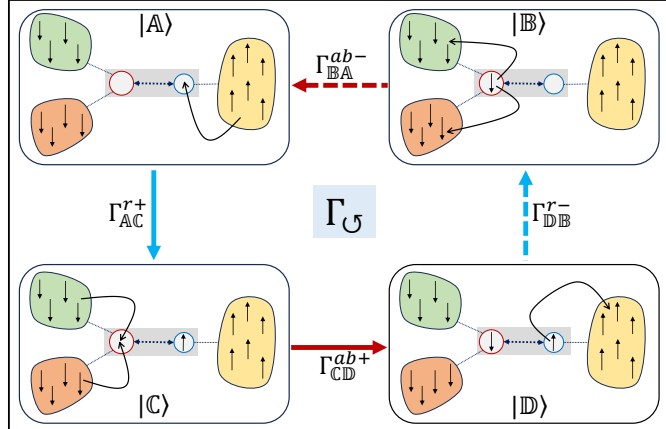

FIG. 2. Schematic representation of the clockwise (Left) and anti-clockwise (Right) transition cycles, induced by energy flow from one bath to another, associated with the particle exchange between QDs and the coupled reservoirs. At steady state, all transition rates of the clockwise (anti-clockwise) cycles are equal to each other and represented by $\Gamma_{\circlearrowright}(\Gamma_{\circlearrowleft})$ [Cf. Eq. (21)].

ated with all three reservoirs, we can write

$$J_{\mathrm{E}}(t) = \sum_{\lambda=a,b,r} J_{\mathrm{E}}^{\lambda}(t) \quad ; \quad J_{\mathrm{N}}(t) = \sum_{\lambda=a,b,r} J_{\mathrm{N}}^{\lambda}(t), \quad (16)$$

where $J_{\mathrm{E(N)}}^{\lambda}$ is positive if energy (particle) current flows from the reservoir $\lambda$ to the system. By comparing Eq.(15) and Eq.(16), we thus find

$$J_{\mathrm{E}}^{\lambda}(t) = \mathrm{Tr}_{\mathrm{s}}[\mathcal{L}_{\lambda}[\rho(t)]H_{\mathrm{s}}] \; ; \; J_{\mathrm{N}}^{\lambda}(t) = \mathrm{Tr}_{\mathrm{s}}[\mathcal{L}_{\lambda}[\rho(t)]\mathcal{N}]. \quad (17)$$

Similarly, we can evaluate the expression of heat current due to the $\lambda$'th reservoir as

$$J_{\mathrm{Q}}^{\lambda}(t) = J_{\mathrm{E}}^{\lambda}(t) - \mu_{\lambda} J_{\mathrm{N}}^{\lambda}(t). \quad (18)$$

Now, at the steady state, both energy and particle currents would be independent of the time, i.e., Eq. (18) reduces to

$$J_{\mathrm{Q}}^{\lambda} = J_{\mathrm{E}}^{\lambda} - \mu_{\lambda} J_{\mathrm{N}}^{\lambda} = \mathrm{Tr}_{\mathrm{s}}[\mathcal{L}_{\lambda}[\rho_{ss}]H_{\mathrm{s}}] - \mu_{\lambda} \mathrm{Tr}_{\mathrm{s}}[\mathcal{L}_{\lambda}[\rho_{ss}]\mathcal{N}]. \quad (19)$$

Using Eq. (4) we obtain the form of $\mathcal{L}_{\lambda}[\rho_{ss}]$, which leads to the formal expressions for the energy, particle, and heat currents as follows [64]

$$J_{\mathrm{E}}^{\lambda} = \sum_{\{\omega_{ij}\}} \omega_{ij}\Gamma_{ij}^{\lambda+} = \sum_{\{\omega_{ji}\}} \omega_{ji}\Gamma_{ji}^{\lambda-};$$

$$J_{\mathrm{N}}^{\lambda} = \sum_{\{\omega_{ij}\}} \Gamma_{ij}^{\lambda+} = \sum_{\{\omega_{ji}\}} \Gamma_{ji}^{\lambda-}; \quad (20)$$

$$J_{\mathrm{Q}}^{\lambda} = \sum_{\{\omega_{ij}\}} (\omega_{ij} - \mu_{\lambda})\Gamma_{ij}^{\lambda+} = \sum_{\{\omega_{ji}\}} (\omega_{ji} - \mu_{\lambda})\Gamma_{ji}^{\lambda-}.$$

The explicit expressions for all three currents are given in Appendix B. At the steady state, the populations of the various eigenstates become time-invariant, i.e., $\dot{\rho}_{\mathbb{A}} = \dot{\rho}_{\mathbb{B}} = \dot{\rho}_{\mathbb{C}} = \dot{\rho}_{\mathbb{D}} = 0$; therefore, we obtain

$$\Gamma_{\mathbb{AB}}^{ab+} = \Gamma_{\mathbb{BD}}^{r+} = \Gamma_{\mathbb{DC}}^{ab-} = \Gamma_{\mathbb{CA}}^{r-} = \Gamma_{\mathbb{ABCDA}} \equiv \Gamma_{\circlearrowright};$$

$$\Gamma_{\mathbb{AC}}^{r+} = \Gamma_{\mathbb{CD}}^{ab+} = \Gamma_{\mathbb{DB}}^{r-} = \Gamma_{\mathbb{BA}}^{ab-} = \Gamma_{\mathbb{ACDBA}} \equiv \Gamma_{\circlearrowleft}, \quad (21)$$

which implies $\Gamma_{\circlearrowright} = -\Gamma_{\circlearrowleft}$ [FIG. 2]. Using the above relations in Eq. (20), one can rewrite the steady state energy and particle currents due to the $r$'th reservoir as

$$J_{\mathrm{E}}^{r} = \varepsilon_{\mathrm{R}}\Gamma_{\circlearrowleft} + (\varepsilon_{\mathrm{R}} + \kappa)\Gamma_{\circlearrowright} = \kappa\Gamma_{\circlearrowright};$$

$$J_{\mathrm{N}}^{r} = \Gamma_{\mathbb{AC}}^{r+} + \Gamma_{\mathbb{BD}}^{r+} = \Gamma_{\circlearrowleft} + \Gamma_{\circlearrowright} = 0. \quad (22)$$

This immediately yields $J_{\mathrm{Q}}^{r} = J_{\mathrm{E}}^{r} = \kappa\Gamma_{\circlearrowright}$. As QD$_{\mathrm{R}}$ is only coupled with lead $r$, there will be no net particle current driven by $r$ at the steady state, i.e., $J_{\mathrm{N}}^{r} = 0$. However, QD$_{\mathrm{L}}$ is simultaneously coupled with leads $a$ and $b$, allowing (spin-polarized) particle flow across $a$ and $b$ at the steady state. There is no external source or sink associated with the system-reservoir model, so the net energy and (spin-polarized) particle currents are conserved in the steady state. To this end, we summarize our first set of analytical results

$$\sum_{\lambda=a,b,r} J_{\mathrm{E}}^{\lambda} = 0, \quad \text{and} \quad \sum_{\lambda=a,b,r} J_{\mathrm{N}}^{\lambda} = 0, \quad (23)$$

which yields

$$J_{\mathrm{E}}^{r} = -J_{\mathrm{E}}^{ab} = \kappa\Gamma_{\circlearrowright}; \qquad J_{\mathrm{N}}^{a} = -J_{\mathrm{N}}^{b}. \quad (24)$$

The explicit form of the $\Gamma_{\circlearrowright}$ is given in Appendix- C]. The heat current is not a conserved quantity which readily follows $J_{\mathrm{Q}}^{a} + J_{\mathrm{Q}}^{b} \equiv J_{\mathrm{Q}}^{ab} \neq J_{\mathrm{E}}^{ab} = -\kappa\Gamma_{\circlearrowright}$. Next, we will discuss how the above steady-state currents are related to the entropy production rate at near equilibrium through force-flux relation.

## V. THERMODYNAMIC FORCE, FLUX AND ENTROPY PRODUCTION

From the definition of the entropy change of the system [Eq. (13)], one can write down the following relation [63]

$$\Delta\mathcal{S}_{\mathrm{s}}(t) = \Sigma(t) + \Phi(t), \quad (25)$$

where the *entropy production* term $\Sigma(t)$ signifies the irreversible contribution of the system entropy change and $\Phi(t)$ term refers to the *entropy flux*, i.e., equivalent to the system entropy change due to the reversible contribution of heat exchanges with the reservoirs. The explicit forms of the $\Sigma(t)$ and the $\Phi(t)$ can be derived [See Appendix- D] following Refs. [62, 63] as:

$$\Sigma(t) = k_B \operatorname{Tr}[\rho_{\text{tot}}(t) \ln\{\rho_{\text{tot}}(t)\}]$$
$$- k_B \operatorname{Tr}\left[\rho_{\text{tot}}(t) \ln\left\{\rho_{\text{s}}(t) \prod_\lambda \rho_\lambda^{\text{eq}}\right\}\right], \quad (26)$$
$$\Phi(t) = k_B \sum_\lambda \operatorname{Tr}_\lambda\left[\{\rho_\lambda(t) - \rho_\lambda^{\text{eq}}\} \ln \rho_\lambda^{\text{eq}}\right],$$

where, $\rho_\lambda^{\text{eq}}$ and $\rho_\lambda(t)$ represent the density operators of the $\lambda$'th reservoir under equilibrium and near equilibrium conditions, respectively. Moreover, $\rho_\lambda^{\text{eq}}$ under grand canonical ensemble, can be written as

$$\rho_\lambda^{\text{eq}} = \frac{e^{-\beta_\lambda H_{\text{B}}^\lambda}}{\mathcal{Z}_\lambda(\beta_\lambda)}, \quad (27)$$

where $H_{\text{B}}^\lambda \equiv H_{\text{B}}^{\lambda\sigma} = \sum_k (\epsilon_k^{\lambda\sigma} - \mu_{\lambda\sigma}) c_{\lambda\sigma k}^\dagger c_{\lambda\sigma k}$ and $\mathcal{Z}_\lambda(\beta_\lambda) = \operatorname{Tr}\left[e^{-\beta_\lambda H_{\text{B}}^\lambda}\right]$ is the the grand canonical partition function for the $\lambda$'th reservoir. As a result, $\Phi(t)$ is identified as the heat exchange with the reservoir between the final and initial time

$$\Phi(t) = -k_B \sum_\lambda \beta_\lambda [\langle H_{\text{B}}^\lambda\rangle_t - \langle H_{\text{B}}^\lambda\rangle_0] = k_B \sum_\lambda \beta_\lambda \Delta Q_\lambda,$$
$$(28)$$

where $\Delta Q_\lambda = \langle H_{\text{B}}^\lambda\rangle_0 - \langle H_{\text{B}}^\lambda\rangle_t$ and $\langle H_{\text{B}}^\lambda\rangle_t = \operatorname{Tr}_\lambda[\rho_\lambda(t) H_{\text{B}}^\lambda]$. Thus, we can rewrite Eq. (25) as

$$\Delta \mathcal{S}_{\text{s}}(t) = \Sigma(t) + k_B \sum_\lambda \beta_\lambda \Delta Q_\lambda, \quad (29)$$

where we replace the second term by Eq. (28). At the steady state, $\dot\Sigma$ becomes minimum [60, 62, 63] and there is no net entropy change in the system, i.e., $\frac{d}{dt}\{\Delta \mathcal{S}_{\text{s}}(t)\} = 0$. Therefore, Eq. (29) yields

$$\dot\Sigma = -k_B \sum_\lambda \beta_\lambda \frac{d}{dt}\{\Delta Q_\lambda\} = -k_B \sum_\lambda \beta_\lambda J_{\text{Q}}^\lambda. \quad (30)$$

Now, inserting the expression for heat current $J_{\text{Q}}^\lambda = J_{\text{E}}^\lambda - \mu_\lambda J_{\text{N}}^\lambda$ in Eq. (30), we obtain

$$\dot\Sigma = -k_B \left[\beta_a J_{\text{E}}^a + \beta_b J_{\text{E}}^b + \beta_r J_{\text{E}}^r\right] + k_B \left[\beta_a \mu_a J_{\text{N}}^a + \beta_b \mu_b J_{\text{N}}^b\right],$$
$$(31)$$

where, we use Eq. (22) for $J_{\text{N}}^r = 0$. At the steady state, since, the overall energy and particle currents are conserved, i.e., $J_{\text{E}} = \sum_\lambda J_{\text{E}}^\lambda = 0$ and $J_{\text{N}} = \sum_\lambda J_{\text{N}}^\lambda = 0$. So, Eq. (31) reduces to

$$\dot\Sigma = J_{\text{E}}^r \mathcal{F}_{\text{E}}^r + J_{\text{E}}^b \mathcal{F}_{\text{E}}^b + J_{\text{N}}^b \mathcal{F}_{\text{N}}^b, \quad (32)$$

where the macroscopic thermodynamic forces conjugate to thermodynamic fluxes $J_{\text{E}}^{r|b}$ and $J_{\text{N}}^b$ are identified as

$$\mathcal{F}_{\text{E}}^{r|b} = k_B(\beta_a - \beta_{r|b}); \quad \mathcal{F}_{\text{N}}^b = k_B(\beta_b \mu_b - \beta_a \mu_a). \quad (33)$$

As a prerequisite, there are now two classes of reduced models that involve coupled transport in energy and (spin-polarized) particle currents:

$(i)$  Reduced model-I ($\mathcal{F}_{\text{E}}^b = 0$): $\dot\Sigma = J_{\text{E}}^r \mathcal{F}_{\text{E}}^r + J_{\text{N}}^b \mathcal{F}_{\text{N}}^b$,

$(ii)$  Reduced model-II ($\mathcal{F}_{\text{E}}^r = 0$): $\dot\Sigma = J_{\text{E}}^b \mathcal{F}_{\text{E}}^b + J_{\text{N}}^b \mathcal{F}_{\text{N}}^b$,

where we set $\beta_a = \beta_b = \beta$ for Reduced model-I [FIG. 1b] and $\beta_a = \beta_r = \beta'$ for Reduced model-II [FIG. 1b]. Note that the above two models are the only possible cases that can be derived from the general model with pairs of thermodynamic forces and fluxes, as required for ICC.

While, Eq. (32) links the entropy production rate to macroscopic forces and fluxes through temperature, chemical potential, and amount of heat exchange with the reservoirs, to grasp the underlying thermodynamic principles of inverse current, we must identify the microscopic expressions of these thermodynamic quantities. The ideal starting point to establish this connection between macroscopic and microscopic thermodynamic frameworks is the entropy production rate. Thus, rewriting the von Neumann entropy defined in Eq. (12)

$$\mathcal{S}_{\text{s}}(t) = -k_B \sum_{\mathbb{i}} \rho_{\mathbb{i}}(t) \ln \rho_{\mathbb{i}}(t), \quad (34)$$

in terms of the microscopic populations of the different system eigenstates, $\{\rho_{\mathbb{i}}\}$, ($\mathbb{i} = \mathbb{A}, \mathbb{B}, \mathbb{C}, \mathbb{D}$), we can evaluate [See Appendix E] the time evolution of the entropy change as [61]

$$\frac{d}{dt}\Delta \mathcal{S}_{\text{s}}(t) = k_B \left[\Gamma_{\mathbb{A}\mathbb{B}}^{a+} \ln\left(\frac{\rho_{\mathbb{A}}}{\rho_{\mathbb{B}}}\right) + \Gamma_{\mathbb{A}\mathbb{B}}^{b+} \ln\left(\frac{\rho_{\mathbb{A}}}{\rho_{\mathbb{B}}}\right) + \Gamma_{\mathbb{B}\mathbb{D}}^{r+} \ln\left(\frac{\rho_{\mathbb{B}}}{\rho_{\mathbb{D}}}\right)\right.$$
$$\left. + \Gamma_{\mathbb{D}\mathbb{C}}^{a-} \ln\left(\frac{\rho_{\mathbb{D}}}{\rho_{\mathbb{C}}}\right) + \Gamma_{\mathbb{D}\mathbb{C}}^{b-} \ln\left(\frac{\rho_{\mathbb{D}}}{\rho_{\mathbb{C}}}\right) + \Gamma_{\mathbb{C}\mathbb{A}}^{r-} \ln\left(\frac{\rho_{\mathbb{C}}}{\rho_{\mathbb{A}}}\right)\right],$$
$$(35)$$

where we use Eq. (7) for $\dot\rho_{\mathbb{i}}(t)$. By comparing the above equation with Eq. (25), we can identify the microscopic version of entropy production rate and entropy flux [58, 61], as follows [Appendix E]

$$\dot{\Sigma}(t) = k_B \left[ (k_{\mathbb{AB}}^{a+}\rho_{\mathbb{A}} - k_{\mathbb{BA}}^{a-}\rho_{\mathbb{B}}) \ln\left( \frac{k_{\mathbb{AB}}^{a+}\rho_{\mathbb{A}}}{k_{\mathbb{BA}}^{a-}\rho_{\mathbb{B}}} \right) + (k_{\mathbb{AB}}^{b+}\rho_{\mathbb{A}} - k_{\mathbb{BA}}^{b-}\rho_{\mathbb{B}}) \ln\left( \frac{k_{\mathbb{AB}}^{b+}\rho_{\mathbb{A}}}{k_{\mathbb{BA}}^{b-}\rho_{\mathbb{B}}} \right) + (k_{\mathbb{BD}}^{r+}\rho_{\mathbb{B}} - k_{\mathbb{DB}}^{r-}\rho_{\mathbb{D}}) \ln\left( \frac{k_{\mathbb{BD}}^{r+}\rho_{\mathbb{B}}}{k_{\mathbb{DB}}^{r-}\rho_{\mathbb{D}}} \right) \right.$$

$$+ (k_{\mathbb{DC}}^{a-}\rho_{\mathbb{D}} - k_{\mathbb{CD}}^{a+}\rho_{\mathbb{C}}) \ln\left( \frac{k_{\mathbb{DC}}^{a-}\rho_{\mathbb{D}}}{k_{\mathbb{CD}}^{a+}\rho_{\mathbb{C}}} \right) + (k_{\mathbb{DC}}^{b-}\rho_{\mathbb{D}} - k_{\mathbb{CD}}^{b+}\rho_{\mathbb{C}}) \ln\left( \frac{k_{\mathbb{DC}}^{b-}\rho_{\mathbb{D}}}{k_{\mathbb{CD}}^{b+}\rho_{\mathbb{C}}} \right) + (k_{\mathbb{CA}}^{r-}\rho_{\mathbb{C}} - k_{\mathbb{AC}}^{r+}\rho_{\mathbb{A}}) \ln\left( \frac{k_{\mathbb{CA}}^{r-}\rho_{\mathbb{C}}}{k_{\mathbb{AC}}^{r+}\rho_{\mathbb{A}}} \right) \left. \right],$$

$$\dot{\Phi}(t) = -k_B \left[ \Gamma_{\mathbb{AB}}^{a+} \ln\left( \frac{k_{\mathbb{AB}}^{a+}}{k_{\mathbb{BA}}^{a-}} \right) + \Gamma_{\mathbb{AB}}^{b+} \ln\left( \frac{k_{\mathbb{AB}}^{b+}}{k_{\mathbb{BA}}^{b-}} \right) + \Gamma_{\mathbb{BD}}^{r+} \ln\left( \frac{k_{\mathbb{BD}}^{r+}}{k_{\mathbb{DB}}^{r-}} \right) + \Gamma_{\mathbb{DC}}^{a-} \ln\left( \frac{k_{\mathbb{DC}}^{a-}}{k_{\mathbb{CD}}^{a+}} \right) + \Gamma_{\mathbb{DC}}^{b-} \ln\left( \frac{k_{\mathbb{DC}}^{b-}}{k_{\mathbb{CD}}^{b+}} \right) + \Gamma_{\mathbb{CA}}^{r-} \ln\left( \frac{k_{\mathbb{CA}}^{r-}}{k_{\mathbb{AC}}^{r+}} \right) \right].$$

$$(36)$$

In Eq. (36), each term of the Schnakenberg entropy production rate $\dot{\Sigma}$ [61], has a form of $(a-b)\ln\left(\frac{a}{b}\right)$, which guarantees the non-negativity of the entropy production rate. At the steady state, it leads to the following form of the entropy production rate [Appendix E]

$$\dot{\Sigma}(t) = -\dot{\Phi}(t) = \kappa \Gamma_{\circlearrowleft} \left[ \left( \frac{k_B}{\kappa} \right) \ln\left( \frac{k_{\mathbb{AB}}^{a+} k_{\mathbb{BD}}^{r+} k_{\mathbb{DC}}^{a-} k_{\mathbb{CA}}^{r-}}{k_{\mathbb{BA}}^{a-} k_{\mathbb{DB}}^{r-} k_{\mathbb{CD}}^{a+} k_{\mathbb{AC}}^{r+}} \right) \right] + \left( \varepsilon_{\mathrm{L}} \Gamma_{\mathbb{AB}}^{b+} - (\varepsilon_{\mathrm{L}} + \kappa) \Gamma_{\mathbb{DC}}^{b-} \right) \left[ \left( \frac{k_B}{\kappa} \right) \ln\left( \frac{k_{\mathbb{BA}}^{b-} k_{\mathbb{AB}}^{a+} k_{\mathbb{CD}}^{b+} k_{\mathbb{DC}}^{a-}}{k_{\mathbb{DC}}^{b-} k_{\mathbb{CD}}^{a+} k_{\mathbb{AB}}^{b+} k_{\mathbb{BA}}^{a-}} \right) \right]$$

$$+ (\Gamma_{\mathbb{AB}}^{b+} - \Gamma_{\mathbb{DC}}^{b-}) \left[ k_B(1+\theta) \ln\left( \frac{k_{\mathbb{AB}}^{b+} k_{\mathbb{BA}}^{a-}}{k_{\mathbb{BA}}^{b-} k_{\mathbb{AB}}^{a+}} \right) + k_B \theta \ln\left( \frac{k_{\mathbb{DC}}^{b-} k_{\mathbb{CD}}^{a+}}{k_{\mathbb{CD}}^{b+} k_{\mathbb{DC}}^{a-}} \right) \right] \equiv J_{\mathrm{E}}^r \mathcal{F}_{\mathrm{E}}^r + J_{\mathrm{E}}^b \mathcal{F}_{\mathrm{E}}^b + J_{\mathrm{N}}^b \mathcal{F}_{\mathrm{N}}^b,$$

$$(37)$$

where $\theta = \left( \frac{\varepsilon_{\mathrm{L}}}{\kappa} \right)$ is the scaled system parameter. The sign of $\theta$ plays a pivotal role in determining the ICC behavior in both energy and (spin-polarized) particle currents, as will be examined in the next section. From the above equation, all three associated fluxes and their conjugated forces are readily identified as follows:

$$J_{\mathrm{E}}^r = \kappa \Gamma_{\circlearrowleft} \quad ; \quad J_{\mathrm{N}}^b = (\Gamma_{\mathbb{AB}}^{b+} - \Gamma_{\mathbb{DC}}^{b-}),$$
$$J_{\mathrm{E}}^b = \varepsilon_{\mathrm{L}} \Gamma_{\mathbb{AB}}^{b+} + (\varepsilon_{\mathrm{L}} + \kappa) \Gamma_{\mathbb{CD}}^{b+} = \varepsilon_{\mathrm{L}} J_{\mathrm{N}}^b + \kappa \Gamma_{\mathbb{CD}}^{b+};$$

$$(38)$$

and

$$\mathcal{F}_{\mathrm{E}}^r = \left( \frac{k_B}{\kappa} \right) \ln\left( \frac{k_{\mathbb{AB}}^{a+} k_{\mathbb{BD}}^{r+} k_{\mathbb{DC}}^{a-} k_{\mathbb{CA}}^{r-}}{k_{\mathbb{BA}}^{a-} k_{\mathbb{DB}}^{r-} k_{\mathbb{CD}}^{a+} k_{\mathbb{AC}}^{r+}} \right) = k_B(\beta_a - \beta_r),$$

$$(39a)$$

$$\mathcal{F}_{\mathrm{E}}^b = \left( \frac{k_B}{\kappa} \right) \ln\left( \frac{k_{\mathbb{BA}}^{b-} k_{\mathbb{AB}}^{a+} k_{\mathbb{CD}}^{b+} k_{\mathbb{DC}}^{a-}}{k_{\mathbb{DC}}^{b-} k_{\mathbb{CD}}^{a+} k_{\mathbb{AB}}^{b+} k_{\mathbb{BA}}^{a-}} \right) = k_B(\beta_a - \beta_b),$$

$$(39b)$$

$$\mathcal{F}_{\mathrm{N}}^b = k_B(\theta + 1) \ln\left( \frac{k_{\mathbb{AB}}^{b+} k_{\mathbb{BA}}^{a-}}{k_{\mathbb{BA}}^{b-} k_{\mathbb{AB}}^{a+}} \right) + k_B \theta \ln\left( \frac{k_{\mathbb{DC}}^{b-} k_{\mathbb{CD}}^{a+}}{k_{\mathbb{CD}}^{b+} k_{\mathbb{DC}}^{a-}} \right)$$
$$= k_B \ln\left( \frac{k_{\mathbb{AB}}^{b+} k_{\mathbb{BA}}^{a-}}{k_{\mathbb{BA}}^{b-} k_{\mathbb{AB}}^{a+}} \right) - \varepsilon_{\mathrm{L}} \mathcal{F}_{\mathrm{E}}^b = k_B(\beta_b \mu_b - \beta_a \mu_a)$$

$$(39c)$$

Equation (38) is identical to the explicit expressions of the thermodynamic fluxes obtained in Sec. IV. The equivalence between the macroscopic and microscopic versions of the above forces [Cf. Eq. (39)] can be verified upon using Eq. (9), followed by inserting the expressions of the

FDFs. In case of $\mathcal{F}_{\mathrm{E}}^b = 0$ (i.e., $\beta_a = \beta_b$), the expression of the particle force $\mathcal{F}_{\mathrm{N}}^b$ reduces to

$$\mathcal{F}_{\mathrm{N}}^b = k_B(\theta + 1) \ln\left( \frac{k_{\mathbb{AB}}^{b+} k_{\mathbb{BA}}^{a-}}{k_{\mathbb{BA}}^{b-} k_{\mathbb{AB}}^{a+}} \right) + k_B \theta \ln\left( \frac{k_{\mathbb{BA}}^{b-} k_{\mathbb{AB}}^{a+}}{k_{\mathbb{AB}}^{b+} k_{\mathbb{BA}}^{a-}} \right)$$
$$= k_B \ln\left( \frac{k_{\mathbb{AB}}^{b+} k_{\mathbb{BA}}^{a-}}{k_{\mathbb{BA}}^{b-} k_{\mathbb{AB}}^{a+}} \right) = k_B \ln\left( \frac{k_{\mathbb{CD}}^{b+} k_{\mathbb{DC}}^{a-}}{k_{\mathbb{DC}}^{b-} k_{\mathbb{CD}}^{a+}} \right) = k_B \beta \Delta \mu.$$

$$(40)$$

Equations (38)-(40) are the first major results of our analysis. Equipped with the above set of Eqs. (37)-(40), we are now ready to explore ICC in the following section.

## VI. RESULT AND DISCUSSION

As outlined in Sec. V, let us analyze the reduced models with two forces and conjugated fluxes, as a precursor of ICC. For all numerical plots, we have considered $\gamma_a = \gamma_b = \gamma_r = \gamma$ and plotted appropriately scaled dimensionless forces and fluxes. The scaled forces are taken either zero or positive without loss of any generality.

### A. Reduced Model-I: ICC in energy current

§ $\mathcal{F}_{\mathrm{E}}^b = 0$ **and** $\dot{\Sigma} = J_{\mathrm{E}}^r \mathcal{F}_{\mathrm{E}}^r + J_{\mathrm{N}}^b \mathcal{F}_{\mathrm{N}}^b$: It is intuitively clear when two forces $(\mathcal{F}_{\mathrm{E}}^r, \mathcal{F}_{\mathrm{N}}^b) = 0$, both currents $(J_{\mathrm{E}}^r, J_{\mathrm{N}}^b)$ would be zero. However, from the viewpoint of the microscopic picture, it provides interesting insights:

If $\mathcal{F}_N^b = 0$, from Eq. (40) we can write

$$\left(\frac{k_{\mathbb{AB}}^{b+}k_{\mathbb{BA}}^{a-}}{k_{\mathbb{BA}}^{b-}k_{\mathbb{AB}}^{a+}}\right) = \left(\frac{k_{\mathbb{CD}}^{b+}k_{\mathbb{DC}}^{a-}}{k_{\mathbb{DC}}^{b-}k_{\mathbb{CD}}^{a+}}\right) = 1, \qquad (41)$$

applying Eq. (10) in the above equation, it can be shown that

$$\left(\frac{k_{\mathbb{AB}}^{b+}}{k_{\mathbb{BA}}^{b-}}\right) = \left(\frac{k_{\mathbb{AB}}^{a+}}{k_{\mathbb{BA}}^{a-}}\right) = \left(\frac{k_{\mathbb{AB}}^{ab+}}{k_{\mathbb{BA}}^{ab-}}\right),$$
$$\left(\frac{k_{\mathbb{CD}}^{b+}}{k_{\mathbb{DC}}^{b-}}\right) = \left(\frac{k_{\mathbb{CD}}^{a+}}{k_{\mathbb{DC}}^{a-}}\right) = \left(\frac{k_{\mathbb{CD}}^{ab+}}{k_{\mathbb{DC}}^{ab-}}\right). \qquad (42)$$

Applying the above relations in Eq. (8), one obtains

$$\Gamma_{\mathbb{AB}}^{ab+} = 2\Gamma_{\mathbb{AB}}^{a+} = 2\Gamma_{\mathbb{AB}}^{b+}; \qquad \Gamma_{\mathbb{CD}}^{ab+} = 2\Gamma_{\mathbb{CD}}^{a+} = 2\Gamma_{\mathbb{CD}}^{b+}, \quad (43)$$

which when combined with Eq. (21), yields $\Gamma_{\mathbb{AB}}^{b+} = -\Gamma_{\mathbb{CD}}^{b+}$. Following Eq. (38), spin-polarized particle current $J_N^b = 0$, irrespective of the value of the non-conjugate force $\mathcal{F}_E^r$ [FIG. 3a: red line]. Thus, Eq. (33) reduces to $\dot{\Sigma} = J_E^r \mathcal{F}_E^r \geq 0$. As a result, $J_E^r$ would always align in the direction of the $\mathcal{F}_E^r$, and it can be verified from the microscopic picture as well. By considering $\beta > \beta_r$ we assume w.l.o.g., $\mathcal{F}_E^r > 0$, such that Eq. (39a) leads to the following relation

$$\left(\frac{k_B}{\kappa}\right) \ln\left(\frac{k_{\mathbb{AB}}^{a+}k_{\mathbb{BD}}^{r+}k_{\mathbb{DC}}^{a-}k_{\mathbb{CA}}^{r-}}{k_{\mathbb{BA}}^{a-}k_{\mathbb{DB}}^{r-}k_{\mathbb{CD}}^{a+}k_{\mathbb{AC}}^{r+}}\right) > 0. \qquad (44)$$

Given, $\mathcal{F}_N^b = 0$, we can utilize Eq. (42) for $\kappa > 0$, to rewrite the above condition as

$$\left(\frac{k_{\mathbb{AB}}^{ab+}\rho_{\mathbb{A}}}{k_{\mathbb{BA}}^{ab-}\rho_{\mathbb{B}}}\right)\left(\frac{k_{\mathbb{BD}}^{r+}\rho_{\mathbb{B}}}{k_{\mathbb{DB}}^{r-}\rho_{\mathbb{D}}}\right)\left(\frac{k_{\mathbb{DC}}^{ab-}\rho_{\mathbb{D}}}{k_{\mathbb{CD}}^{ab+}\rho_{\mathbb{C}}}\right)\left(\frac{k_{\mathbb{CA}}^{r-}\rho_{\mathbb{C}}}{k_{\mathbb{AC}}^{r+}\rho_{\mathbb{A}}}\right) > 1. \quad (45)$$

To satisfy the above condition, at steady state, each term in the parenthesis must be greater than 1. Applying this in Eqs. (8) and (21), we obtain, $\Gamma_{\circlearrowright} > 0$. Similarly, for $\kappa < 0$, we deduce $\Gamma_{\circlearrowright} < 0$. In both cases, the energy current is defined as $J_E^r = \kappa\Gamma_{\circlearrowright} > 0$ [FIG. 3a: blue line]. So, ICC can occur neither in energy nor (spin-polarized) particle currents.

• **When** $\mathcal{F}_E^r = 0$: The non-negativity of the entropy production rate $\dot{\Sigma} = J_N^b \mathcal{F}_N^b \geq 0$, implies that $J_N^b$ would always align in the direction of $\mathcal{F}_N^b$ [FIG. 3a: red line]. This can be verified also from the microscopic picture. Without loss of any generality, we consider $\mathcal{F}_N^b > 0$ in Eq. (40) which yields

$$\left(\frac{k_{\mathbb{AB}}^{b+}k_{\mathbb{BA}}^{a-}}{k_{\mathbb{BA}}^{b-}k_{\mathbb{AB}}^{a+}}\right) > 1 \quad ; \quad \left(\frac{k_{\mathbb{CD}}^{b+}k_{\mathbb{DC}}^{a-}}{k_{\mathbb{DC}}^{b-}k_{\mathbb{CD}}^{a+}}\right) > 1, \qquad (46)$$

and immediately suggests

$$\left(\frac{k_{\mathbb{AB}}^{b+}}{k_{\mathbb{BA}}^{b-}}\right) > \left(\frac{k_{\mathbb{AB}}^{a+}}{k_{\mathbb{BA}}^{a-}}\right) \quad ; \quad \left(\frac{k_{\mathbb{CD}}^{b+}}{k_{\mathbb{DC}}^{b-}}\right) > \left(\frac{k_{\mathbb{CD}}^{a+}}{k_{\mathbb{DC}}^{a-}}\right). \qquad (47)$$

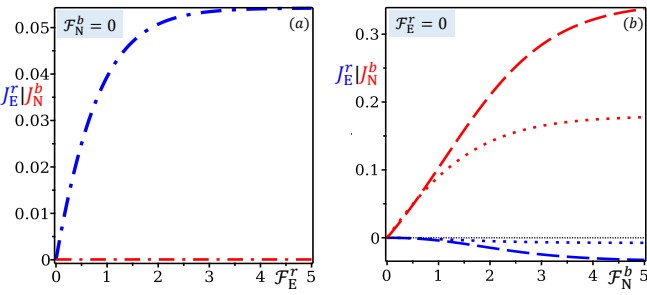

FIG. 3. Variation of $J_E^r$ (blue line) and $J_N^b$ (red line) against **(a)** $\mathcal{F}_E^r$ while $\mathcal{F}_N^b = 0$ i.e. $(\mu_a = \mu_b)$ and **(b)** $\mathcal{F}_N^b$ while $\mathcal{F}_E^r = 0$ i.e. $(\beta = \beta_r)$. For all cases, the (dashed) dotted lines correspond to $\kappa = (-)+1.5\hbar\gamma$, validating the condition $|\kappa| > \varepsilon_L$. Other system and bath parameters: $\varepsilon_L = 1.0\hbar\gamma$, $\varepsilon_R = 2.5\hbar\gamma$, $\beta_r = 1/\hbar\gamma$ and $\mu_b = 1.0\hbar\gamma$.

Applying Eq. (10) in Eq. (47), it can be shown that, $k_{\mathbb{BA}|\mathbb{DC}}^{a-} > k_{\mathbb{BA}|\mathbb{DC}}^{b-}$ and $k_{\mathbb{AB}|\mathbb{CD}}^{a+} < k_{\mathbb{AB}|\mathbb{CD}}^{b+}$. Using these relations in Eq. (8) we obtain, $\Gamma_{\mathbb{AB}|\mathbb{CD}}^{b+} > \Gamma_{\mathbb{AB}|\mathbb{CD}}^{a+}$. Now, we can define two variables X and Y as

$$X = \Gamma_{\mathbb{AB}}^{b+} - \Gamma_{\mathbb{AB}}^{a+} \quad ; \quad Y = \Gamma_{\mathbb{CD}}^{b+} - \Gamma_{\mathbb{CD}}^{a+}, \qquad (48)$$

such that we can write

$$\Gamma_{\mathbb{AB}}^{ab+} = 2\Gamma_{\mathbb{AB}}^{b+} - X \quad ; \quad \Gamma_{\mathbb{CD}}^{ab+} = 2\Gamma_{\mathbb{CD}}^{b+} - Y. \qquad (49)$$

Using Eqs. (21) and (38), it's straightforward to demonstrate from Eq. (49) that under steady-state conditions, $J_N^b = \frac{1}{2}(X+Y) > 0$, as both $X$ and $Y$ are positive in this case. Again, from Eq. (39a) it follows

$$\left(\frac{k_{\mathbb{AB}}^{a+}k_{\mathbb{BD}}^{r+}k_{\mathbb{DC}}^{a-}k_{\mathbb{CA}}^{r-}}{k_{\mathbb{BA}}^{a-}k_{\mathbb{DB}}^{r-}k_{\mathbb{CD}}^{a+}k_{\mathbb{AC}}^{r+}}\right) = 1. \qquad (50)$$

We can rearrange the above conditions as

$$\left(\frac{k_{\mathbb{AB}}^{ab+}k_{\mathbb{BD}}^{r+}k_{\mathbb{DC}}^{ab-}k_{\mathbb{CA}}^{r-}}{k_{\mathbb{BA}}^{ab-}k_{\mathbb{DB}}^{r-}k_{\mathbb{CD}}^{ab+}k_{\mathbb{AC}}^{r+}}\right) = \left(\frac{k_{\mathbb{BA}}^{a+}k_{\mathbb{AB}}^{ab+}k_{\mathbb{CD}}^{a+}k_{\mathbb{DC}}^{ab-}}{k_{\mathbb{AB}}^{a-}k_{\mathbb{BA}}^{ab-}k_{\mathbb{DC}}^{a-}k_{\mathbb{CD}}^{ab+}}\right), \qquad (51)$$

where the r.h.s can be re-written as

$$\text{r.h.s} = \left\{\frac{1 + (k_{\mathbb{AB}}^{b+}/k_{\mathbb{AB}}^{a+})}{1 + (k_{\mathbb{BA}}^{b-}/k_{\mathbb{BA}}^{a-})}\right\}\left\{\frac{1 + (k_{\mathbb{DC}}^{b-}/k_{\mathbb{DC}}^{a-})}{1 + (k_{\mathbb{CD}}^{b+}/k_{\mathbb{CD}}^{a+})}\right\} \equiv \mathcal{P}\mathcal{Q}. \qquad (52)$$

It is trivial to show that

$$(k_{\mathbb{AB}}^{b+}/k_{\mathbb{AB}}^{a+}) > 1 \quad \text{and} \quad (k_{\mathbb{BA}}^{b-}/k_{\mathbb{BA}}^{a-}) < 1 \quad \Rightarrow \mathcal{P} > 1;$$
$$(k_{\mathbb{DC}}^{b-}/k_{\mathbb{DC}}^{a-}) < 1 \quad \text{and} \quad (k_{\mathbb{CD}}^{b+}/k_{\mathbb{CD}}^{a+}) > 1 \quad \Rightarrow \mathcal{Q} < 1, (53)$$

hence, we obtain from Eq. (52) that $\mathcal{P}\mathcal{Q} \gtrless 1$. Further, it is possible to verify

$$\left\{\frac{1 + (k_{\mathbb{AB}}^{b+}/k_{\mathbb{AB}}^{a+})}{1 + (k_{\mathbb{BA}}^{b-}/k_{\mathbb{BA}}^{a-})}\right\} < \left\{\frac{1 + (k_{\mathbb{CD}}^{b-}/k_{\mathbb{CD}}^{a-})}{1 + (k_{\mathbb{DC}}^{b+}/k_{\mathbb{DC}}^{a+})}\right\}, \qquad (54)$$

under the conditions $\mathcal{F}_E^b \geqslant 0$ and $\kappa > 0$, which gives $\mathcal{PQ} < 1$. Thus, r.h.s of Eq. (51) reduces to

$$\left(\frac{k_{AB}^{ab+}k_{BD}^{r+}k_{DC}^{ab-}k_{CA}^{r-}}{k_{BA}^{ab-}k_{DB}^{r-}k_{CD}^{ab+}k_{AC}^{r+}}\right) = \mathcal{PQ} < 1. \tag{55}$$

For $\kappa < 0$, following similar arguments, one can easily show that $\mathcal{PQ} > 1$. Hence, following the same treatment done in Eqs. (44) (45), one may conclude from Eqs. (8) and (21) that $\Gamma_{\circlearrowleft} \lessgtr 0$ for $\kappa \gtrless 0$, respectively. This immediately suggests that the energy current, $J_E^r = \kappa\Gamma_{\circlearrowleft} < 0$, irrespective of the sign of $\kappa$ [FIG. 3b: blue line]. Thus, we have an energy current flowing against $\mathcal{F}_N^b$ as *pseudo-ICC*, a special type of cross-effect.

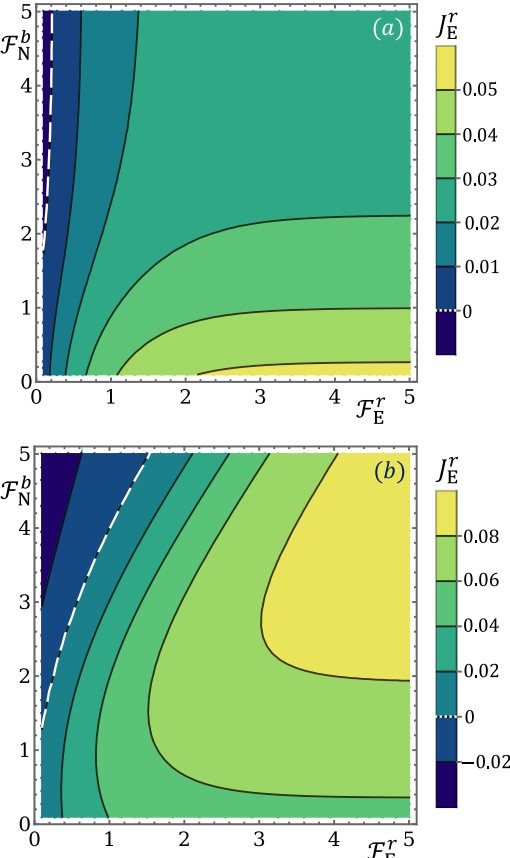

FIG. 4. Variation of the energy current $J_E^r$ with both thermodynamic forces $\mathcal{F}_E^r$ and $\mathcal{F}_N^b$ for positive and negative $\kappa$: **(a)** $\kappa = 1.5\hbar\gamma$ and **(b)** $\kappa = -1.5\hbar\gamma$. For both plots, the region left to the white dashed line signifies the genuine ICC in the energy current. Other system and bath parameters: $\varepsilon_L = 1.0\hbar\gamma$, $\varepsilon_R = 2.5\hbar\gamma$, $\beta_r = 1/\hbar\gamma$, and $\mu_b = 1.0\hbar\gamma$.

A note on the cross-effect and the term pseudo-ICC is appropriate here. In both cases, one force is set to zero, causing the conjugate flux to be unaffected by its corresponding force, which lacks a specific direction. Under these conditions, the current can flow either along or against the non-conjugate force, as both are thermo-

dynamically allowed. If the flux aligns with the non-conjugate force, we call it as the *normal* cross-effect or simply "cross-effect". If the flux opposes the applied force, we classify it as a *special* type of cross-effect or pseudo-ICC. While achieving genuine ICC is a challenging task, pseudo-ICC could serve as an initial indication. In the Reduced model-I, since pseudo-ICC behavior is observed solely in the energy current, true ICC may only be expected in the energy current.

• **When** $(\mathcal{F}_E^r, \mathcal{F}_N^b) > 0$: Among the four possibilities, both fluxes could align in the direction of their conjugate force, validating the positivity of the entropy production rate $\dot{\Sigma} = J_E^r\mathcal{F}_E^r + J_N^b\mathcal{F}_N^b$. However, both fluxes simultaneously can't operate against both forces, which would otherwise violate the laws of thermodynamics. Therefore, we can have ICC in either $J_E^r$ or $J_N^b$. However, ICC in the $J_N^b$ is not possible in this reduced model-I, as discussed above. Hence, we will focus on the true ICC effect in $J_E^r$ only.

It is trivial to understand when $J_E^r$ flows against both of the forces, the model behaves like an autonomous refrigerator that generates energy current against the thermal gradient. The other flux $J_N^b$ must be positive and compensate the negative contribution of $J_E^r$ to make the entropy production rate non-negative. Given the microscopic condition of $\mathcal{F}_N^b > 0$, we have already verified that $J_N^b > 0$. Again, $\mathcal{F}_E^r > 0$, so in view of Eqs. (39a), the microscopic condition described by Eq. (50), is modified as

$$\left(\frac{k_{AB}^{a+}k_{BD}^{r+}k_{DC}^{a-}k_{CA}^{r-}}{k_{BA}^{a-}k_{DB}^{r-}k_{CD}^{a+}k_{AC}^{r+}}\right) \gtrless 1, \quad \text{for} \quad \kappa \gtrless 0. \tag{56}$$

Following the similar argument mentioned in Eqs. (50)-(52), we can rewrite the above condition as

$$\left(\frac{k_{AB}^{ab+}k_{BD}^{r+}k_{DC}^{ab-}k_{CA}^{r-}}{k_{BA}^{ab-}k_{DB}^{r-}k_{CD}^{ab+}k_{AC}^{r+}}\right) \gtrless \mathcal{PQ} \quad \text{for} \quad \kappa \gtrless 0. \tag{57}$$

We have already shown as long as $\mathcal{F}_E^b \geqslant 0$, one can write $\mathcal{PQ} \lessgtr 1$ for $\kappa \gtrless 0$, respectively. Hence, the final condition from Eq. (57) reduces to

$$\left(\frac{k_{AB}^{ab+}k_{BD}^{r+}k_{DC}^{ab-}k_{CA}^{r-}}{k_{BA}^{ab-}k_{DB}^{r-}k_{CD}^{ab+}k_{AC}^{r+}}\right) \gtrless 1, \tag{58}$$

which is not governed by the sign of $\kappa$. Now with a similar procedure presented before, Eq. (58) can result in $\Gamma_{\circlearrowleft} \gtrless 0$ for any values of $\kappa$. Hence, the energy current $J_E^r = \kappa\Gamma_{\circlearrowleft}$, could be positive or negative, for $\kappa \gtrless 0$. While $J_E^r > 0$ is normally expected as both forces $(\mathcal{F}_E^r, \mathcal{F}_N^b) > 0$, but, $J_E^r < 0$, signifies genuine ICC, as the energy current would flow against both the forces, indicating the autonomous refrigeration effect, which can be obtained in both positive and negative $\kappa$. It is evident, to maintain a positive entropy production rate, ICC in energy current is achieved only for a small $\mathcal{F}_E^r$ and large $\mathcal{F}_N^b$, as depicted in FIG. 4. Thus, we can implement our Reduced model-I as an autonomous refrigeration device [32, 37, 65–68]

i.e., a spin-thermoelectric machine that drives the energy current from the cold reservoir to the hot reservoir by the influence of the spin-polarized particle current. The Co-efficient of performance (COP) of such device is given by [16, 17]

$$\zeta = \left| \frac{J_Q^r}{\Delta\mu \cdot J_N^b} \right|, \tag{59}$$

where the numerator, $|J_Q^r| = |J_E^r|$ represents the cooling power.

## B. Reduced Model-II: ICC in both energy and (spin-polarized) particle currents

§ $\mathcal{F}_E^r = 0$ **and** $\dot{\Sigma} = J_E^b \mathcal{F}_E^b + J_N^b \mathcal{F}_N^b$**:** It is apparent that there would be no flux when both forces are zero. To explore pseudo-ICC as a primary criterion for ICC, we proceed by setting one non-zero force at a time. Subsequently, we will set both forces non-zero to ascertain genuine ICC.

(i) **When** $\mathcal{F}_N^b = 0$ **and** $\mathcal{F}_E^b > 0$**:** Upon setting conditions in Eq. (39c) for the particle force, one can obtain after a little bit of algebra

$$\ln\left(\frac{k_{\mathbb{AB}}^{b+}k_{\mathbb{BA}}^{a-}}{k_{\mathbb{BA}}^{b-}k_{\mathbb{AB}}^{a+}}\right) = \theta\ln\left(\frac{k_{\mathbb{BA}}^{b-}k_{\mathbb{AB}}^{a+}k_{\mathbb{CD}}^{b+}k_{\mathbb{DC}}^{a-}}{k_{\mathbb{DC}}^{b-}k_{\mathbb{CD}}^{a+}k_{\mathbb{AB}}^{b+}k_{\mathbb{BA}}^{a-}}\right) > 0,$$

or, $(1+\theta)\ln\left(\frac{k_{\mathbb{AB}}^{b+}k_{\mathbb{BA}}^{a-}}{k_{\mathbb{BA}}^{b-}k_{\mathbb{AB}}^{a+}}\right) = \theta\ln\left(\frac{k_{\mathbb{CD}}^{b+}k_{\mathbb{DC}}^{a-}}{k_{\mathbb{DC}}^{b-}k_{\mathbb{CD}}^{a+}}\right) > 0,$ (60)

or, $\ln\left(\frac{k_{\mathbb{AB}}^{b+}k_{\mathbb{BA}}^{a-}}{k_{\mathbb{BA}}^{b-}k_{\mathbb{AB}}^{a+}}\right) = \left(\frac{\theta}{1+\theta}\right)\ln\left(\frac{k_{\mathbb{CD}}^{b+}k_{\mathbb{DC}}^{a-}}{k_{\mathbb{DC}}^{b-}k_{\mathbb{CD}}^{a+}}\right) > 0,$

where, $\frac{\theta}{1+\theta} = \frac{\varepsilon_L}{\varepsilon_L+\kappa}$. So, there are two possibilities:

● For $\kappa > 0$, it directly reduces to Eq. (46), with both X and Y positive, which finally leads to the condition $J_N^b > 0$. Hence, the spin-induced particle current is always positive [FIG. 5a], indicating the presence of the normal cross-effect, as the conjugate force $\mathcal{F}_N^b$ is zero.

● If $\kappa < 0$ and $|\kappa| > \varepsilon_L$, the positions of the energy eigenstates $|\mathbb{C}\rangle$ and $|\mathbb{D}\rangle$ are interchanged, and the Eq. (60) reduces to

$$\left(\frac{k_{\mathbb{AB}}^{b+}k_{\mathbb{BA}}^{a-}}{k_{\mathbb{BA}}^{b-}k_{\mathbb{AB}}^{a+}}\right) > 1 \quad ; \quad \left(\frac{k_{\mathbb{CD}}^{b+}k_{\mathbb{DC}}^{a-}}{k_{\mathbb{DC}}^{b-}k_{\mathbb{CD}}^{a+}}\right) < 1. \tag{61}$$

Following a similar procedure presented in the context of Eqs. (46) (49), one can show that $J_N^b = \frac{1}{2}(X + Y) \gtrless 0$, as X is always positive and Y is always negative following Eq. (61). Hence, $J_N^b$ could be either positive or negative, indicating the possibility of both the cross-effect and pseudo-ICC [FIG. 5a], given that the conjugate force $\mathcal{F}_N^b = 0$. As a result, $\dot{\Sigma} = J_E^b \mathcal{F}_E^b$ and $J_E^b$ aligns with the direction of $\mathcal{F}_E^b$.

In summary, when $\mathcal{F}_E^b$ is the only force acting on the system, $J_E^b$ is always positive [FIG. 5a]. If $\kappa > 0$, $J_N^b$ is

positive, indicating the cross-effect. Conversely, if $\kappa < 0$, $J_N^b$ can be either positive or negative, suggesting the possibility of true ICC in the spin-induced particle current. While Eq. (46) indicates that both $|\mathbb{A}\rangle \to |\mathbb{B}\rangle$ and $|\mathbb{C}\rangle \to |\mathbb{D}\rangle$ transitions are primarily controlled by the same reservoir (bath $b$), Eq. (61) signifies a competition between the two coupled reservoirs (baths $b$ and $a$) in controlling the respective transitions. This competition between the two reservoirs results in $J_N^b$ being negative.

(ii) **When** $\mathcal{F}_E^b = 0$ **and** $\mathcal{F}_N^b > 0$**:** In this case, $J_N^b$ is always positive [FIG. 5b], in confirmation with the non-negativity of the entropy production rate $\dot{\Sigma} = J_N^b \mathcal{F}_N^b$. However, the energy current $J_E^b$ is strongly dependent on the sign of $\kappa$ and can be expressed as $J_E^b > \kappa\Gamma_{\mathbb{CD}}^{b+}$ [Cf. Eq.(38)]. As demonstrated in Eqs.(50)-(55), when $\mathcal{F}_E^r = 0$, the value of $\mathcal{PQ} \lessgtr 1$ under the conditions $\mathcal{F}_E^b = 0$ and $\kappa \gtrless 0$. This implies

$$\left(\frac{k_{\mathbb{AB}}^{ab+}k_{\mathbb{BD}}^{r+}k_{\mathbb{DC}}^{ab-}k_{\mathbb{CA}}^{r-}}{k_{\mathbb{BA}}^{ab-}k_{\mathbb{DB}}^{r-}k_{\mathbb{CD}}^{ab+}k_{\mathbb{AC}}^{r+}}\right) \lessgtr 1. \tag{62}$$

Utilizing the steady state condition [Cf. Eq. (21)], we can derive from Eq. (62)

$$\left(\frac{k_{\mathbb{DC}}^{ab-} \rho_{\mathbb{D}}}{k_{\mathbb{CD}}^{ab+} \rho_{\mathbb{C}}}\right) \lessgtr 1. \tag{63}$$

The above equation suggests that $\Gamma_{\mathbb{CD}}^{ab+} \gtrless 0$. Substituting this into Eq. (49), we obtain $\Gamma_{\mathbb{CD}}^{b+} \gtrless \frac{Y}{2}$ for $\kappa \gtrless 0$. This implies that $\Gamma_{\mathbb{CD}}^{b+}$ is always positive when $\kappa > 0$, but it can be positive or negative for $\kappa < 0$. Therefore, the energy current $J_E^b$ is always positive for $\kappa > 0$, as $J_E^b > \kappa\Gamma_{\mathbb{CD}}^{b+}$. However, for $\kappa < 0$, the quantity $\kappa\Gamma_{\mathbb{CD}}^{b+}$ can be either positive or negative, resulting in $J_E^b$ being either positive or negative [FIG. 5b]. To summarize, when $\mathcal{F}_N^b$ is the only

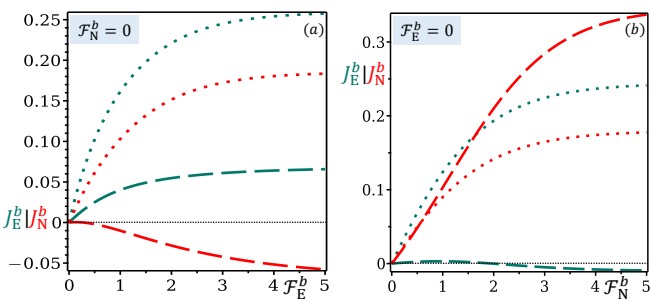

FIG. 5. Variation of the currents $J_E^b$ (bluegreen line) and $J_N^b$ (red line) against the thermodynamic forces **(a)** $\mathcal{F}_E^b$ while $\mathcal{F}_N^b = 0$ i.e. $(\mu_a\beta' = \mu_b\beta_b)$ and **(b)** $\mathcal{F}_N^b$ while $\mathcal{F}_E^b = 0$ i.e. $(\beta' = \beta_b)$. For all cases, the (dashed) dotted lines correspond to $\kappa = (-) + 1.5\hbar\gamma$, validating the condition $|\kappa| > \varepsilon_L$. Other system and bath parameters: $\varepsilon_L = 1.0\hbar\gamma$, $\varepsilon_R = 2.5\hbar\gamma$, $\beta_b = 1/\hbar\gamma$, $\mu_b = 1.0\hbar\gamma$.

non-zero force acting on the system, the spin-induced particle current $J_N^b$ is always positive. The energy current $J_E^b$ is positive only if $\kappa > 0$, while it can be either positive

or negative for $\kappa < 0$. $J_{\mathrm{E}}^b > 0$ represents the cross-effect, whereas $J_{\mathrm{E}}^b < 0$ signifies the pseudo-ICC in the energy current, as the current flows against the non-conjugate particle force $\mathcal{F}_{\mathrm{N}}^b$. Hence, there is a possibility of true ICC occurring in the energy current as well within this reduced model.

(iii) **When both** $(\mathcal{F}_{\mathrm{E}}^b, \mathcal{F}_{\mathrm{N}}^b) > 0$: From our discussion of the previous two cases, we have found that pseudo-ICC arises in energy and particle currents only when $\kappa < 0$. Thus, we will focus solely on the $\kappa < 0$ regime.

• **ICC in particle current:** Since $\mathcal{F}_{\mathrm{N}}^b > 0$, it readily follows from Eq. (39c)

$$\ln\left(\frac{\mathrm{k}_{\mathbb{AB}}^{b+}\mathrm{k}_{\mathbb{BA}}^{a-}}{\mathrm{k}_{\mathbb{BA}}^{b-}\mathrm{k}_{\mathbb{AB}}^{a+}}\right) > \left(\frac{\varepsilon_{\mathrm{L}}}{k_B}\right)\mathcal{F}_{\mathrm{E}}^b > 0. \tag{64}$$

Inserting the expression of $\mathcal{F}_{\mathrm{E}}^b$ from Eq. (39b), in the above equation, we obtain

$$\ln\left(\frac{\mathrm{k}_{\mathbb{AB}}^{b+}\mathrm{k}_{\mathbb{BA}}^{a-}}{\mathrm{k}_{\mathbb{BA}}^{b-}\mathrm{k}_{\mathbb{AB}}^{a+}}\right) > \left(\frac{\varepsilon_{\mathrm{L}}}{\varepsilon_{\mathrm{L}} + \kappa}\right)\ln\left(\frac{\mathrm{k}_{\mathbb{CD}}^{b+}\mathrm{k}_{\mathbb{DC}}^{a-}}{\mathrm{k}_{\mathbb{DC}}^{b-}\mathrm{k}_{\mathbb{CD}}^{a+}}\right), \tag{65}$$

which then implies

$$\left(\frac{\mathrm{k}_{\mathbb{AB}}^{b+}\mathrm{k}_{\mathbb{BA}}^{a-}}{\mathrm{k}_{\mathbb{BA}}^{b-}\mathrm{k}_{\mathbb{AB}}^{a+}}\right) > 1; \quad \left(\frac{\mathrm{k}_{\mathbb{CD}}^{b+}\mathrm{k}_{\mathbb{DC}}^{a-}}{\mathrm{k}_{\mathbb{DC}}^{b-}\mathrm{k}_{\mathbb{CD}}^{a+}}\right) \gtrless 1. \tag{66}$$

So, we can continue our discussion by classifying the above relations into two categories: (i) In the first case, both of the arguments of the above relations are greater than 1, and (ii) in the second case, the latter argument is considered to be less than 1. For case (i), following the same treatment done in Eq. (46), it can be shown as a trivial case that $J_{\mathrm{N}}^b = \frac{1}{2}(\mathrm{X}+\mathrm{Y}) > 0$, as both X and Y are positive, i.e., the spin-induced particle current would be positive as long as the conjugate force $\mathcal{F}_{\mathrm{N}}^b$ is positive. For case (ii), as the latter argument is less than 1, Y would be negative. Again, following the same method done in case of Eq. (61), we can obtain, $J_{\mathrm{N}}^b = \frac{1}{2}(\mathrm{X}+\mathrm{Y}) \gtrless 0$, as X is positive and Y is negative. So, we can have genuine ICC in the particle current, defined by $J_{\mathrm{N}}^b < 0$ [FIG. 6a], which indicates the flow of the spin-induced particle current against both the non-zero positive forces $\mathcal{F}_{\mathrm{N}}^b$ and $\mathcal{F}_{\mathrm{E}}^b$. Thus, our Reduced model-II can be implemented as the spin-thermoelectric heat engine [31, 33, 69, 70], which drives the spin-polarized particle flux against the particle force, influenced by the non-conjugate energy force $\mathcal{F}_{\mathrm{E}}^b$. Finally, one can define the output power as $|\Delta\mu \cdot J_{\mathrm{N}}^b|$, so that the efficiency of the device is given by [16, 17]

$$\eta = \left|\frac{\Delta\mu \cdot J_{\mathrm{N}}^b}{J_{\mathrm{Q}}^r}\right|. \tag{67}$$

• **ICC in energy current:** Under the first case when both the arguments of Eq. (66) are greater than 1, the spin-induced particle current $J_{\mathrm{N}}^b > 0$. As a consequence, we obtain from Eq. (38), $J_{\mathrm{E}}^b > \kappa\Gamma_{\mathbb{CD}}^{b+}$. Again, $\mathcal{F}_{\mathrm{E}}^r$ is anyway zero, and under the conditions $\mathcal{F}_{\mathrm{E}}^b > 0$ and $\kappa < 0$,

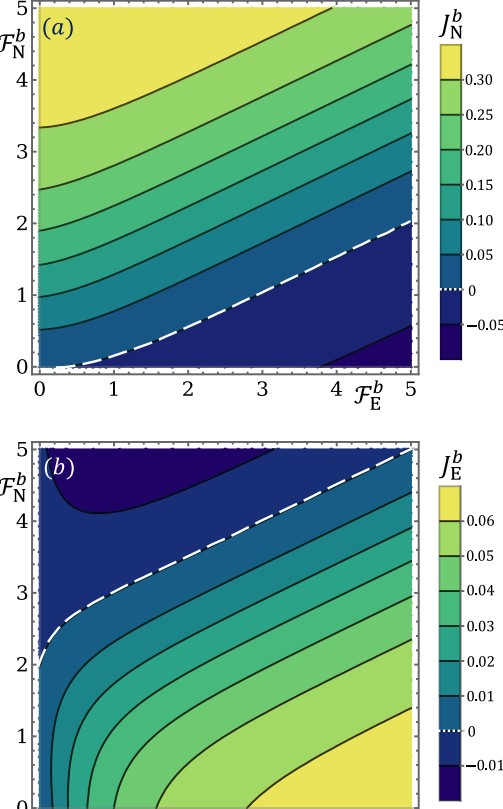

FIG. 6. Variation of **(a)** the spin-polarised particle current $J_{\mathrm{N}}^b$ and **(b)** the energy current $J_{\mathrm{E}}^r$ with both the thermodynamic forces for negative $\kappa$ i.e. $\kappa = -1.5\hbar\gamma$. Genuine ICC for $J_{\mathrm{N}}^b$ occurs in the area below the white dashed line in (a) while it occurs for $J_{\mathrm{E}}^r$ above the white dashed line in (b). Other system and bath parameters: $\varepsilon_{\mathrm{L}} = 1.0\hbar\gamma$, $\varepsilon_{\mathrm{R}} = 2.5\hbar\gamma$, $\beta_b = 1/\hbar\gamma$, $\mu_b = 1.0\hbar\gamma$.

we have already shown in Eq. (50)- (55) that $\mathcal{P}\mathcal{Q} > 1$, which results in

$$\left(\frac{\mathrm{k}_{\mathbb{AB}}^{ab+}\mathrm{k}_{\mathbb{BD}}^{r+}\mathrm{k}_{\mathbb{DC}}^{ab-}\mathrm{k}_{\mathbb{CA}}^{r-}}{\mathrm{k}_{\mathbb{BA}}^{ab-}\mathrm{k}_{\mathbb{DB}}^{r-}\mathrm{k}_{\mathbb{CD}}^{ab+}\mathrm{k}_{\mathbb{AC}}^{r+}}\right) > 1. \tag{68}$$

From the above relation, we have previously established in Eq. (62) that $\Gamma_{\mathbb{CD}}^{ab+} < 0$ for the $\kappa < 0$, which finally leads to $\Gamma_{\mathbb{CD}}^{b+} < \frac{\mathrm{Y}}{2}$, where Y is positive. Thus, $\Gamma_{\mathbb{CD}}^{b+}$ can be both positive or negative. As a result, energy flux $J_{\mathrm{E}}^b$ defined as $J_{\mathrm{E}}^b > \kappa\Gamma_{\mathbb{CD}}^{b+}$, can be positive or negative, i.e., $J_{\mathrm{E}}^b \gtrless 0$. The genuine ICC will occur when $J_{\mathrm{E}}^b < 0$ [FIG. (6)b], indicating the flow of energy current against both non-zero positive forces $\mathcal{F}_{\mathrm{N}}^b$ and $\mathcal{F}_{\mathrm{E}}^b$. Consequently, the Reduced model-II functions as a thermoelectric refrigerator [32, 37, 65–68], propelling energy flux against the temperature gradient, influenced by the non-conjugate particle force $\mathcal{F}_{\mathrm{N}}^b$. The COP for such a device is given by Eq. (59).

So, we conclude that when both forces of the Reduced model-II, $\mathcal{F}_{\mathrm{E}}^b$ and $\mathcal{F}_{\mathrm{N}}^b$ are positive, genuine ICC can be

achieved in both energy and spin-induced particle currents. Thus, our model can be implemented both as a spin-thermoelectric heat engine and a refrigerator. It is evident that true ICC in the spin-induced particle current $J_{\mathrm{N}}^{b}$ is obtained when both arguments of Eq. (66) are positive and $\kappa < 0$, whereas true ICC in the energy current $J_{\mathrm{E}}^{b}$ is achieved, when the latter argument of Eq. (66) is negative, under the restriction $-\kappa > \varepsilon_{\mathrm{L}} > 0$. These results complement the fact that two ICC regions do not overlap with each other [FIG. 6] in compliance with the second law of thermodynamics and positivity of the entropy production rate. It is worth mentioning that genuine ICC can be explored in both energy and particle currents within the general model as well, under various system-bath parameters. However, conducting analytical studies based on microscopic descriptions will be challenging for the general model.

## VII.  CONCLUSIONS

Identification of exact thermodynamic forces and conjugate fluxes in the presence of multiple reservoirs is crucial for any thermodynamic description of novel phenomena. Here we present a comprehensive quantum thermodynamic theory of the inverse current phenomenon in coupled transport, where one induced current opposes all the thermodynamic forces present in the system. Based on a simple variant of Sánchez-Büttiker model of a three-terminal Coulomb-coupled quantum dots, we have examined the counter-intuitive inverse current behavior observed in energy and spin-induced particle currents near equilibrium situations. Our analysis focuses on the macroscopic and microscopic correspondence of the entropy production rate, employing the grand-canonical formalism of the Lindblad master equation and the Schnakenberg entropy formulation. The linearity of the quantum master equation allows exact analytical expressions for thermodynamic forces and fluxes, incorporating both macroscopic reservoir parameters and microscopic system characteristics. It enables us to uniquely identify all thermodynamic force-flux pairs for both general and reduced models, facilitating a systematic analysis of genuine ICC behavior in both energy and spin-polarized particle currents. Finally, we illustrate that our model can function as an autonomous spin-thermoelectric engine or refrigerator by exploiting ICC in spin-induced particle currents and energy currents, respectively. While we demonstrate an autonomous quantum-dot refrigerator is easier to achieve, where chemical work done by current-carrying quantum particles assists in transferring thermal energy from a cold to a hot bath, autonomous engine realization necessitates attractive interaction between coupled dots. We anticipate that our findings will be valuable in developing ICC-assisted unconventional spin-thermometric quantum dot devices in the near future.

### ACKNOWLEDGEMENTS

AG acknowledges IITK for infrastructure and financial support. S.G. acknowledges the Ministry of Education, Government of India, for the Prime Minister Research Fellowship (PMRF). N.G. is thankful to CSIR for the fellowship.

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

## Appendix A: Derivation of the LME

We present the interaction picture master equation to study the system's dynamical evolution that exchanges particles and energy with the reservoirs. Let us start with the tunneling Hamiltonian

$$H_{\mathrm{T}} = H_{\mathrm{T}}^{\mathrm{L}\downarrow \mathrm{a}} + H_{\mathrm{T}}^{\mathrm{L}\downarrow \mathrm{b}} + H_{\mathrm{T}}^{\mathrm{R}\uparrow \mathrm{r}} \tag{A1}$$

where, $H_{\mathrm{T}}^{\mathrm{L}\downarrow \mathrm{a(b)}} = \hbar \sum_k [t_k^{\mathrm{L}\downarrow \mathrm{a(b)}} c_{a(b)\downarrow k}^{\dagger} d_{\mathrm{L}\downarrow} + t_k^{\mathrm{L}\downarrow \mathrm{a(b)}*} d_{\mathrm{L}\downarrow}^{\dagger} c_{a(b)\downarrow k}]$ and $H_{\mathrm{T}}^{\mathrm{R}\uparrow \mathrm{r}} = \hbar \sum_k [t_k^{\mathrm{R}\uparrow \mathrm{r}} c_{r\uparrow k}^{\dagger} d_{\mathrm{R}\uparrow} + t_k^{\mathrm{R}\uparrow \mathrm{r}*} d_{\mathrm{R}\uparrow}^{\dagger} c_{r\uparrow k}]$. To derive the master equation, we start with the von Neumann equation for the total density matrix $\rho_{\mathrm{tot}}$ in the interaction picture

$$\frac{d}{dt}\rho_{\mathrm{tot}} = -\frac{i}{\hbar}[H_{\mathrm{T}}(t), \rho_{\mathrm{tot}}(t)]. \tag{A2}$$

Integrating the above equation, and taking a trace over the bath degrees of freedom, one obtains

$$\frac{\partial}{\partial t}\rho_{\mathrm{s}}(t) = \frac{1}{(i\hbar)^2} \int_0^t dt' \mathrm{Tr}_{a,b,r}[H_{\mathrm{T}}(t), [H_{\mathrm{T}}(t-t'), \rho_{\mathrm{tot}}(t')]], \tag{A3}$$

where, $\mathrm{Tr}_{a,b,r}$ refers to the trace over each bath degrees of freedom and $\mathrm{Tr}_{a,b,r}\{\rho_{\mathrm{tot}}(t)\} = \rho_{\mathrm{s}}(t)$ denotes the reduced density operator for the system. We also assume that $\mathrm{Tr}_{a,b,r}[H_{\mathrm{T}}(t), \rho_{\mathrm{tot}}(0)] = 0$. Under the Born-Markov approximation, the above equation can be rewritten as [48, 59, 71]

$$\dot{\rho}_{\mathrm{s}}(t) = \frac{1}{(i\hbar)^2} \sum_{\lambda=a,b,r} \int_0^{\infty} dt' \ \mathrm{Tr}_{a,b,r}[H_{\mathrm{T}}^{\lambda}(t), [H_{\mathrm{T}}^{\lambda}(t-t'), \rho_{\mathrm{s}}(t) \otimes \rho_a \otimes \rho_b \otimes \rho_r]], \tag{A4}$$

where we use the following properties of the bath operators $\mathrm{Tr}_{\lambda}\{c_{\lambda}(t)\rho_{\lambda}\} = 0 = \mathrm{Tr}_{\lambda}\{c_{\lambda}^{\dagger}(t)\rho_{\lambda}\}$ and $\mathrm{Tr}_{a,b,r}\{[H_{\mathrm{T}}^{\lambda}(t), [H_{\mathrm{T}}^{\nu}(t-t'), \rho_{\mathrm{s}}(t) \otimes \rho_a \otimes \rho_b \otimes \rho_r]]\} = 0; \lambda \neq \nu; \lambda, \nu = a, b, r$. Now, in the above equation, we use the interaction picture system and bath operators

$$d_{\alpha\sigma}(t) = e^{iH_{\mathrm{s}}t/\hbar} d_{\alpha\sigma} e^{-iH_{\mathrm{s}}t/\hbar} = \sum_{\omega_{ij}>0} e^{-i\omega_{ij}t/\hbar} d_{\alpha\sigma}; \qquad \alpha\sigma = \mathrm{L}\downarrow, \mathrm{R}\uparrow$$

$$c_{\lambda\sigma}(t) = e^{iH_{\mathrm{B}}t/\hbar} c_{\lambda\sigma} e^{-iH_{\mathrm{B}}t/\hbar} = \sum_k e^{-i(\epsilon_k^{\lambda\sigma}-\mu_\lambda)t/\hbar} c_{\lambda\sigma}; \qquad \lambda\sigma = a\downarrow, b\downarrow, r\uparrow \tag{A5}$$

and their hermitian adjoints, where $\omega_{ij}$ is defined as the transition energy for the transition between the system eigenstates $|i\rangle$ and $|j\rangle$. Eliminating the high-frequency oscillating terms by the standard procedure of secular approximation, one can finally derive the master equation in the following form

$$\dot{\rho}_{\mathrm{s}}(t) = \mathcal{L}_a[\rho_{\mathrm{s}}(t)] + \mathcal{L}_b[\rho_{\mathrm{s}}(t)] + \mathcal{L}_r[\rho_{\mathrm{s}}(t)], \tag{A6}$$

where the Lindblad operators $\mathcal{L}_\lambda[\rho_{\mathrm{s}}(t)]$ are given by

$$\mathcal{L}_\lambda[\rho_{\mathrm{s}}(t)] = \sum_{\{\omega_\alpha\}>0} \mathcal{G}_\lambda(\omega_\alpha) \left[ d_{\alpha\sigma}^{\dagger}(\omega_\alpha)\rho_{\mathrm{s}} d_{\alpha\sigma}(\omega_\alpha) - \frac{1}{2}\{\rho_{\mathrm{s}}, d_{\alpha\sigma}(\omega_\alpha)d_{\alpha\sigma}^{\dagger}(\omega_\alpha)\} \right]$$

$$+ \mathcal{G}_\lambda(-\omega_\alpha) \left[ d_{\alpha\sigma}(\omega_\alpha)\rho_{\mathrm{s}} d_{\alpha\sigma}^{\dagger}(\omega_\alpha) - \frac{1}{2}\{\rho_{\mathrm{s}}, d_{\alpha\sigma}^{\dagger}(\omega_\alpha)d_{\alpha\sigma}(\omega_\alpha)\} \right]. \tag{A7}$$

In the above equation, we define the temperature-dependent bath spectral functions as

$$\mathcal{G}_\lambda(\omega_\alpha) = \gamma_\lambda(\omega_\alpha)f_\lambda^+(\omega_\alpha); \quad \mathcal{G}_\lambda(-\omega_\alpha) = \gamma_\lambda(\omega_\alpha)f_\lambda^-(\omega_\alpha). \tag{A8}$$

The function $f_\lambda^{\pm}(\omega_{ij})$ represents the Fermi distribution functions (FDF) which are obtained by tracing over the bath density operator, for example, $f_{\lambda\sigma}^+(\omega_{ij}) = \mathrm{Tr}_{\lambda\sigma}\left(c_{\lambda\sigma}^{\dagger}c_{\lambda\sigma}\rho_{\lambda\sigma}\right)$, and $f_{\lambda\sigma}^-(\omega_{ij}) = \mathrm{Tr}_{\lambda\sigma}\left(c_{\lambda\sigma}c_{\lambda\sigma}^{\dagger}\rho_{\lambda\sigma}\right)$, where the bath

operators $c_{\lambda\sigma}^{\dagger}$ and $c_{\lambda\sigma}$ obey anti-commutation relation and the reservoir governs the transition between eigenstate $|\mathbb{i}\rangle$ to $|\mathbb{j}\rangle$, that costs $\omega_{\mathbb{ij}}$ amount of energy. The explicit expressions of the FDFs for various transitions are listed below:

$$f_{a|b}^{+}(\omega_{\mathbb{AB}}) = \left[1 + \exp\left(\frac{\varepsilon_{\mathrm{L}} - \mu_{a|b}}{k_B T_{a|b}}\right)\right]^{-1} \equiv f_{a|b}^{1+} \quad , \quad f_{a|b}^{+}(\omega_{\mathbb{CD}}) = \left[1 + \exp\left(\frac{\varepsilon_{\mathrm{L}} + \kappa - \mu_{a|b}}{k_B T_{a|b}}\right)\right]^{-1} \equiv f_{a|b}^{2+},$$

$$f_{a|b}^{-}(\omega_{\mathbb{BA}}) = \left[1 + \exp\left(\frac{-\varepsilon_{\mathrm{L}} + \mu_{a|b}}{k_B T_{a|b}}\right)\right]^{-1} \equiv f_{a|b}^{1-} \quad , \quad f_{a|b}^{-}(\omega_{\mathbb{CD}}) = \left[1 + \exp\left(\frac{-\varepsilon_{\mathrm{L}} - \kappa + \mu_{a|b}}{k_B T_{a|b}}\right)\right]^{-1} \equiv f_{a|b}^{2-},$$

$$f_{r}^{+}(\omega_{\mathbb{AC}}) = \left[1 + \exp\left(\frac{\varepsilon_{\mathrm{R}} - \mu_{r}}{k_B T_{r}}\right)\right]^{-1} \equiv f_{r}^{1+} \quad , \quad f_{r}^{+}(\omega_{\mathbb{BD}}) = \left[1 + \exp\left(\frac{\varepsilon_{\mathrm{R}} + \kappa - \mu_{r}}{k_B T_{r}}\right)\right]^{-1} \equiv f_{r}^{2+},$$

$$f_{r}^{-}(\omega_{\mathbb{CA}}) = \left[1 + \exp\left(\frac{-\varepsilon_{\mathrm{R}} + \mu_{r}}{k_B T_{r}}\right)\right]^{-1} \equiv f_{r}^{1-} \quad , \quad f_{r}^{-}(\omega_{\mathbb{DB}}) = \left[1 + \exp\left(\frac{-\varepsilon_{\mathrm{R}} - \kappa + \mu_{r}}{k_B T_{r}}\right)\right]^{-1} \equiv f_{r}^{2-}. \tag{A9}$$

## Appendix B: Explicit expressions of various currents

From Eq. (20), one can evaluate the explicit expressions of both energy and particle currents for each reservoir in the following way

$$J_{\mathrm{E}}^{a} = \omega_{\mathbb{AB}}\Gamma_{\mathbb{AB}}^{a+} + \omega_{\mathbb{CD}}\Gamma_{\mathbb{CD}}^{a+} = \varepsilon_{\mathrm{L}}\Gamma_{\mathbb{AB}}^{a+} + (\varepsilon_{\mathrm{L}} + \kappa)\Gamma_{\mathbb{CD}}^{a+}; \quad J_{\mathrm{E}}^{b} = \omega_{\mathbb{AB}}\Gamma_{\mathbb{AB}}^{b+} + \omega_{\mathbb{CD}}\Gamma_{\mathbb{CD}}^{b+} = \varepsilon_{\mathrm{L}}\Gamma_{\mathbb{AB}}^{b+} + (\varepsilon_{\mathrm{L}} + \kappa)\Gamma_{\mathbb{CD}}^{b+};$$
$$J_{\mathrm{E}}^{r} = \omega_{\mathbb{AC}}\Gamma_{\mathbb{AC}}^{r+} + \omega_{\mathbb{BD}}\Gamma_{\mathbb{BD}}^{r+} = \varepsilon_{\mathrm{R}}\Gamma_{\mathbb{AC}}^{r+} + (\varepsilon_{\mathrm{R}} + \kappa)\Gamma_{\mathbb{BD}}^{r+}; \quad J_{\mathrm{N}}^{a} = \Gamma_{\mathbb{AB}}^{a+} + \Gamma_{\mathbb{CD}}^{a+}; J_{\mathrm{N}}^{b} = \Gamma_{\mathbb{AB}}^{b+} + \Gamma_{\mathbb{CD}}^{b+}; J_{\mathrm{N}}^{r} = \Gamma_{\mathbb{AC}}^{r+} + \Gamma_{\mathbb{BD}}^{r+}. \tag{B1}$$

Inserting the above relations, the expression of the heat current associated with each reservoir can be evaluated as

$$J_{\mathrm{Q}}^{a} = (\varepsilon_{\mathrm{L}} - \mu_a)\Gamma_{\mathbb{AB}}^{a+} + (\varepsilon_{\mathrm{L}} + \kappa - \mu_a)\Gamma_{\mathbb{CD}}^{a+}; \quad J_{\mathrm{Q}}^{b} = (\varepsilon_{\mathrm{L}} - \mu_b)\Gamma_{\mathbb{AB}}^{b+} + (\varepsilon_{\mathrm{L}} + \kappa - \mu_b)\Gamma_{\mathbb{CD}}^{b+}; \quad J_{\mathrm{Q}}^{r} = (\varepsilon_{\mathrm{R}} - \mu_r)\Gamma_{\mathbb{AC}}^{r+} + (\varepsilon_{\mathrm{R}} + \kappa - \mu_r)\Gamma_{\mathbb{BD}}^{r+}. \tag{B2}$$

## Appendix C: Expression of the steady state transition rate

To determine the steady-state transition rate, we rewrite Eq. (7)-(9) as

$$\mathcal{M} \begin{bmatrix} \rho_{\mathbb{A}} \\ \rho_{\mathbb{B}} \\ \rho_{\mathbb{C}} \\ \rho_{\mathbb{D}} \end{bmatrix} = \begin{bmatrix} 0 \\ 0 \\ 0 \\ 1 \end{bmatrix}, \tag{C1}$$

subject to the condition $\rho_{\mathbb{A}} + \rho_{\mathbb{B}} + \rho_{\mathbb{C}} + \rho_{\mathbb{D}} = 1$, and

$$\mathcal{M} = \begin{bmatrix} -f_r^{1+} - f_a^{1+} - f_b^{1+} & f_a^{1-} + f_b^{1-} & f_r^{1+} & 0 \\ f_a^{1+} + f_b^{1+} & -f_a^{1-} - f_b^{1-} - f_r^{2+} & 0 & f_r^{2-} \\ f_r^{1+} & 0 & -f_a^{2+} - f_b^{2+} - f_r^{1-} & f_a^{2-} + f_b^{2-} \\ 1 & 1 & 1 & 1 \end{bmatrix}, \tag{C2}$$

where, for the sake of simplicity of our analysis, we assume that $\gamma_a \simeq \gamma_b \simeq \gamma_r \equiv \gamma$. Solving Eq. (C1) with the above matrix $\mathcal{M}$, one can obtain the steady state population $\{\rho_{\mathbb{i}}\}$ in terms of which we can evaluate the explicit expression

$$\Gamma_{\circlearrowright} = -\Gamma_{\circlearrowleft} = \gamma\left[\frac{f_{ab}^{1+}[f_{ab}^{2+}(f_r^{2+} - f_r^{1+}) + 2f_r^{2+}(f_r^{1+} - 1)] - 2f_r^{1+}f_{ab}^{2+}(f_r^{2+} - 1)}{3f_{ab}^{1+}(f_r^{1+} - f_r^{2+}) - 6 + 3f_{ab}^{2+}(f_r^{2+} - f_r^{1+})}\right], \tag{C3}$$

where, we define $f_{ab}^{1+(2+)} = f_a^{1+(2+)} + f_b^{1+(2+)}$.

## Appendix D: Non-negativity of entropy production rate

One can evaluate the expression of the entropy production ($\Sigma$) from the entropy change of the system ($\Delta \mathcal{S}_{\mathrm{s}} = \mathcal{S}_{\mathrm{s}}(t) - \mathcal{S}_{\mathrm{s}}(0)$), which is defined as

$$\Delta \mathcal{S}_{\mathrm{s}}(t) = -k_B \operatorname{Tr}_{\mathrm{s}}[\rho_{\mathrm{s}}(t) \ln \rho_{\mathrm{s}}(t)] + k_B \operatorname{Tr}_{\mathrm{s}}[\rho_{\mathrm{s}}(0) \ln \rho_{\mathrm{s}}(0)] = -k_B \operatorname{Tr}[\rho_{\mathrm{tot}}(t) \ln \rho_{\mathrm{s}}(t)] + k_B \operatorname{Tr}[\rho_{\mathrm{tot}}(0) \ln \rho_{\mathrm{s}}(0)]. \tag{D1}$$

We assume that the initial equilibrium state, $\rho_{\text{tot}}(0)$ does not display any entanglement or correlation between the system and the environment. Therefore

$$\rho_{\text{tot}}(0) = \rho_{\text{s}}(0) \prod_{\lambda} \rho_{\lambda}^{\text{eq}}. \tag{D2}$$

Inserting Eq. (D2) into Eq. (D1), we can continue as follows

$$\Delta \mathcal{S}_{\text{s}}(t) = -k_B \operatorname{Tr}[\rho_{\text{tot}}(t) \ln \rho_{\text{s}}(t)] + k_B \operatorname{Tr}[\rho_{\text{tot}}(0) \ln \rho_{\text{tot}}(0)] - k_B \sum_{\lambda} \operatorname{Tr}[\rho_{\text{tot}}(0) \ln \rho_{\lambda}^{\text{eq}}]$$

$$= -k_B \operatorname{Tr}\left[\rho_{\text{tot}}(t) \ln \left\{\rho_{\text{s}}(t) \prod_{\lambda} \rho_{\lambda}^{\text{eq}}\right\}\right] + k_B \operatorname{Tr}[\rho_{\text{tot}}(0) \ln \rho_{\text{tot}}(0)] + k_B \sum_{\lambda} \operatorname{Tr}_{\lambda}[\{\rho_{\lambda}(t) - \rho_{\lambda}^{\text{eq}}\} \ln \rho_{\lambda}^{\text{eq}}]. \tag{D3}$$

Again, $\rho_{\text{tot}}(t)$ and $\rho_{\text{tot}}(0)$ are related through unitary evolution $\rho_{\text{tot}}(t) = \mathcal{U}\rho_{\text{tot}}(0)\mathcal{U}^{\dagger}$, which implies $\operatorname{Tr}[\rho_{\text{tot}}(t) \ln \rho_{\text{tot}}(t)] = \operatorname{Tr}[\rho_{\text{tot}}(0) \ln \rho_{\text{tot}}(0)]$. Applying this relation in the above equation, the final expression of the entropy change of the system can be calculated as

$$\Delta \mathcal{S}_{\text{s}}(t) = -k_B \operatorname{Tr}\left[\rho_{\text{tot}}(t) \ln \left\{\rho_{\text{s}}(t) \prod_{\lambda} \rho_{\lambda}^{\text{eq}}\right\}\right] + k_B \operatorname{Tr}[\rho_{\text{tot}}(t) \ln \rho_{\text{tot}}(t)] + k_B \sum_{\lambda} \operatorname{Tr}_{\lambda}[\{\rho_{\lambda}(t) - \rho_{\lambda}^{\text{eq}}\} \ln \rho_{\lambda}^{\text{eq}}]. \tag{D4}$$

The last term of the above equation can be identified as the *entropy flow* ($\Phi$), representing the reversible contribution to the system entropy change due to heat exchange with the reservoirs. A comparison of the above equation with Eq. (25), defines the *entropy production*, representing the irreversible contribution to the entropy change of the system and the *entropy flow* as

$$\Sigma(t) = k_B \operatorname{Tr}[\rho_{\text{tot}}(t) \ln\{\rho_{\text{tot}}(t)\}] - k_B \operatorname{Tr}\left[\rho_{\text{tot}}(t) \ln \left\{\rho_{\text{s}}(t) \prod_{\lambda} \rho_{\lambda}^{\text{eq}}\right\}\right]; \qquad \Phi(t) = k_B \sum_{\lambda} \operatorname{Tr}_{\lambda} \left[\{\rho_{\lambda}(t) - \rho_{\lambda}^{\text{eq}}\} \ln \rho_{\lambda}^{\text{eq}}\right]. \tag{D5}$$

The above expression of the *entropy production* can be expressed in terms of the relative entropy

$$\Sigma(t) \equiv \mathcal{D}\left[\rho_{\text{tot}}(t) || \left\{\rho_{\text{s}}(t) \prod_{\lambda} \rho_{\lambda}^{\text{eq}}\right\}\right], \tag{D6}$$

where $\mathcal{D}[\rho||\rho']$ is the quantum relative entropy between two density matrices $\rho$ and $\rho'$, defined via

$$\mathcal{D}[\rho||\rho'] := \operatorname{Tr}[\rho \ln \rho] - \operatorname{Tr}[\rho \ln \rho']. \tag{D7}$$

The non-negativity of relative entropy is affirmed, attaining a value of zero solely in the case of complete matrix identity. While the non-negativity of entropy production doesn't imply the same for its rate, in the limit of large reservoirs, $\Sigma(t)$ is expected to converge to a convex, monotonically increasing function of time [62]. In the same limit, if the system dynamics are described by a Markovian quantum Lindblad master equation, implying entropy production as a convex functional of the system density matrix [59, 62], the entropy production rate $\dot{\Sigma}(t)$ would eventually be positive, only reaching zero for the equilibrium state.

## Appendix E: Derivation of the microscopic definition of the entropy production rate

Considering the von Neumann entropy defined as $\mathcal{S}_{\text{s}}(t) = -k_B \sum_{\mathbb{i}} \rho_{\mathbb{i}}(t) \ln \rho_{\mathbb{i}}(t)$, where, $\rho_{\mathbb{i}}$ signifies populations of the system eigenstates ($\mathbb{i} = \mathbb{A}, \mathbb{B}, \mathbb{C}, \mathbb{D}$), the change in the system entropy is given by

$$\Delta \mathcal{S}_{\text{s}}(t) = \mathcal{S}_{\text{s}}(t) - \mathcal{S}_{\text{s}}(0) = -k_B \sum_{\mathbb{i}} \rho_{\mathbb{i}}(t) \ln \rho_{\mathbb{i}}(t) + k_B \sum_{\mathbb{i}} \rho_{\mathbb{i}}(0) \ln \rho_{\mathbb{i}}(0). \tag{E1}$$

So, the time evolution of the entropy change can be evaluated as

$$\frac{d}{dt} \Delta \mathcal{S}_{\text{s}}(t) = -k_B \sum_{\mathbb{i}} \dot{\rho}_{\mathbb{i}}(t) \ln \rho_{\mathbb{i}}(t) + k_B \sum_{\mathbb{i}} \dot{\rho}_{\mathbb{i}}(0) \ln \rho_{\mathbb{i}}(0) = -k_B \sum_{\mathbb{i}} \dot{\rho}_{\mathbb{i}}(t) \ln \rho_{\mathbb{i}}(t) \equiv -k_B \sum_{\mathbb{i}} \dot{\rho}_{\mathbb{i}} \ln \rho_{\mathbb{i}}. \tag{E2}$$

Now, using Eq. (7) for $\dot\rho_i$, we recover Eq. (35) of the main text:

$$\frac{d}{dt}\Delta\mathcal{S}_\mathrm{s}(t) = k_B\left[\Gamma^{a+}_{AB}\ln\left(\frac{\rho_A}{\rho_B}\right) + \Gamma^{b+}_{AB}\ln\left(\frac{\rho_A}{\rho_B}\right) + \Gamma^{r+}_{BD}\ln\left(\frac{\rho_B}{\rho_D}\right) + \Gamma^{a-}_{DC}\ln\left(\frac{\rho_D}{\rho_C}\right) + \Gamma^{b-}_{DC}\ln\left(\frac{\rho_D}{\rho_C}\right) + \Gamma^{r-}_{CA}\ln\left(\frac{\rho_C}{\rho_A}\right)\right]. \quad (E3)$$

By comparing the above equation with Eq. (25), we obtain Eq. (36) for the general expressions of $\dot\Sigma(t)$ and $\dot\Phi(t)$:

$$\dot\Sigma(t) = k_B\left[\Gamma^{a+}_{AB}\ln\left(\frac{k^{a+}_{AB}\rho_A}{k^{a-}_{BA}\rho_B}\right) + \Gamma^{b+}_{AB}\ln\left(\frac{k^{b+}_{AB}\rho_A}{k^{b-}_{BA}\rho_B}\right) + \Gamma^{r+}_{BD}\ln\left(\frac{k^{r+}_{BD}\rho_B}{k^{r-}_{DB}\rho_D}\right) + \Gamma^{a-}_{DC}\ln\left(\frac{k^{a-}_{DC}\rho_D}{k^{a+}_{CD}\rho_C}\right) + \Gamma^{b-}_{DC}\ln\left(\frac{k^{b-}_{DC}\rho_D}{k^{b+}_{CD}\rho_C}\right) + \Gamma^{r-}_{CA}\ln\left(\frac{k^{r-}_{CA}\rho_C}{k^{r+}_{AC}\rho_A}\right)\right]$$

$$= k_B\left[(k^{a+}_{AB}\rho_A - k^{a-}_{BA}\rho_B)\ln\left(\frac{k^{a+}_{AB}\rho_A}{k^{a-}_{BA}\rho_B}\right) + (k^{b+}_{AB}\rho_A - k^{b-}_{BA}\rho_B)\ln\left(\frac{k^{b+}_{AB}\rho_A}{k^{b-}_{BA}\rho_B}\right) + (k^{r+}_{BD}\rho_B - k^{r-}_{DB}\rho_D)\ln\left(\frac{k^{r+}_{BD}\rho_B}{k^{r-}_{DB}\rho_D}\right)\right.$$

$$\left. + (k^{a-}_{DC}\rho_D - k^{a+}_{CD}\rho_C)\ln\left(\frac{k^{a-}_{DC}\rho_D}{k^{a+}_{CD}\rho_C}\right) + (k^{b-}_{DC}\rho_D - k^{b+}_{CD}\rho_C)\ln\left(\frac{k^{b-}_{DC}\rho_D}{k^{b+}_{CD}\rho_C}\right) + (k^{r-}_{CA}\rho_C - k^{r+}_{AC}\rho_A)\ln\left(\frac{k^{r-}_{CA}\rho_C}{k^{r+}_{AC}\rho_A}\right)\right],$$

$$\dot\Phi(t) = -k_B\left[\Gamma^{a+}_{AB}\ln\left(\frac{k^{a+}_{AB}}{k^{a-}_{BA}}\right) + \Gamma^{b+}_{AB}\ln\left(\frac{k^{b+}_{AB}}{k^{b-}_{BA}}\right) + \Gamma^{r+}_{BD}\ln\left(\frac{k^{r+}_{BD}}{k^{r-}_{DB}}\right) + \Gamma^{a-}_{DC}\ln\left(\frac{k^{a-}_{DC}}{k^{a+}_{CD}}\right) + \Gamma^{b-}_{DC}\ln\left(\frac{k^{b-}_{DC}}{k^{b+}_{CD}}\right) + \Gamma^{r-}_{CA}\ln\left(\frac{k^{r-}_{CA}}{k^{r+}_{AC}}\right)\right].$$

$$(E4)$$

There is no net entropy change in the system at the steady state, which reduces Eq. (37) for the form of the entropy production rate:

$$\dot\Sigma(t) = -\dot\Phi(t) = k_B\left[\Gamma^{a+}_{AB}\ln\left(\frac{k^{a+}_{AB}}{k^{a-}_{BA}}\right) + \Gamma^{b+}_{AB}\ln\left(\frac{k^{b+}_{AB}}{k^{b-}_{BA}}\right) + \Gamma^{r+}_{BD}\ln\left(\frac{k^{r+}_{BD}}{k^{r-}_{DB}}\right) + \Gamma^{a-}_{DC}\ln\left(\frac{k^{a-}_{DC}}{k^{a+}_{CD}}\right) + \Gamma^{b-}_{DC}\ln\left(\frac{k^{b-}_{DC}}{k^{b+}_{CD}}\right) + \Gamma^{r-}_{CA}\ln\left(\frac{k^{r-}_{CA}}{k^{r+}_{AC}}\right)\right]$$

$$= k_B\left[\Gamma^{ab+}_{AB}\ln\left(\frac{k^{a+}_{AB}}{k^{a-}_{BA}}\right) + \Gamma^{r+}_{BD}\ln\left(\frac{k^{r+}_{BD}}{k^{r-}_{DB}}\right) + \Gamma^{ab-}_{DC}\ln\left(\frac{k^{a-}_{DC}}{k^{a+}_{CD}}\right) + \Gamma^{r-}_{CA}\ln\left(\frac{k^{r-}_{CA}}{k^{r+}_{AC}}\right)\right] + k_B\left[\Gamma^{b+}_{AB}\ln\left(\frac{k^{b+}_{AB}k^{a-}_{BA}}{k^{b-}_{BA}k^{a+}_{AB}}\right) + \Gamma^{b-}_{DC}\ln\left(\frac{k^{b-}_{DC}k^{a+}_{CD}}{k^{b+}_{CD}k^{a-}_{DC}}\right)\right]$$

$$= k_B\Gamma_\circlearrowright\ln\left(\frac{k^{a+}_{AB}k^{r+}_{BD}k^{a-}_{DC}k^{r-}_{CA}}{k^{a-}_{BA}k^{r-}_{DB}k^{a+}_{CD}k^{r+}_{AC}}\right) + k_B(\Gamma^{b+}_{AB} - \Gamma^{b-}_{DC})\ln\left(\frac{k^{b+}_{AB}k^{a-}_{BA}}{k^{b-}_{BA}k^{a+}_{AB}}\right) + k_B\Gamma^{b-}_{DC}\ln\left(\frac{k^{b-}_{DC}k^{a+}_{CD}k^{b+}_{AB}k^{a-}_{BA}}{k^{b-}_{BA}k^{a+}_{AB}k^{b+}_{CD}k^{a-}_{DC}}\right)$$

$$= \kappa\Gamma_\circlearrowright\left[\left(\frac{k_B}{\kappa}\right)\ln\left(\frac{k^{a+}_{AB}k^{r+}_{BD}k^{a-}_{DC}k^{r-}_{CA}}{k^{a-}_{BA}k^{r-}_{DB}k^{a+}_{CD}k^{r+}_{AC}}\right)\right] + \{\varepsilon_\mathrm{L}\Gamma^{b+}_{AB} - (\varepsilon_\mathrm{L} + \kappa)\Gamma^{b-}_{DC}\}\left[\left(\frac{k_B}{\kappa}\right)\ln\left(\frac{k^{b-}_{BA}k^{a+}_{AB}k^{b+}_{CD}k^{a-}_{DC}}{k^{b-}_{DC}k^{a+}_{CD}k^{b+}_{AB}k^{a-}_{BA}}\right)\right]$$

$$+ (\Gamma^{b+}_{AB} - \Gamma^{b-}_{DC})\left[k_B(1+\theta)\ln\left(\frac{k^{b+}_{AB}k^{a-}_{BA}}{k^{b-}_{BA}k^{a+}_{AB}}\right) + k_B\theta\ln\left(\frac{k^{b-}_{DC}k^{a+}_{CD}}{k^{b+}_{CD}k^{a-}_{DC}}\right)\right].$$

$$(E5)$$