# Peer review of "Inverse Current in Coupled Transport: A Quantum Thermodynamic Framework for Energy and Spin-polarized Particle Currents"

_SciPost Physics_

## Round 2 · Referee Report · Anonymous (Referee 1) · 2025-5-4

Strengths

It is hard to say that this stage (see my report).

Weaknesses

I believe that this manuscript is simply confused, and is giving a NEW name to OLD physics, making it unsuitable for publication.
(see my report).

Report

My recommendation to SciPost is to NOT publish this manuscript as it is. This work was previously submitted to Phys. Rev. B, where I was a referee, and I recommended not publishing it there. While there have been some changes to the manuscript since that submission, they do NOT address the CENTRAL points that led me to reject the manuscript from Phys. Rev. B.

To explain my opinion to SciPost's editors, I now reproduce the pertinent points from my previous 1st and 2nd round reports to Phys. Rev. B (my 2nd round report mentioned agreement with the other referee's 1st round report, see below). The authors have (of course) already seen all these point.

I think the authors are very confused about thermoelectric responses. They claim their "Inverse Current in Coupled Transport (ICC)" are different from well-known thermoelectric effects. Yet their definition makes ICCs exactly the SAME as very well-known negative thermoelectric responses (those with negative Seebeck or Peltier coefficients). So I believe that this manuscript is simply confused, and is giving a NEW name to OLD physics, making it unsuitable for publication.

I note that the other referee (referee 1 for Phys. Rev. B) was already of this opinion in their 1st round report for Phys Rev B, where they wrote that such as the ICC effect "is not counterintuitive, at least to a reader with a basic knowledge of physics. Indeed, every refrigerator works in that way." At that moment, I had a doubt about this, because I was not sure exactly what the authors were claiming. However, the author's responses to our first round reports clarified their claims, and I realized that the other referee was 100% correct.

I recommend that the authors work to better understand thermoelectric effects in their three-terminal two-dot system. Such effects in that system have been extensively studied, both theory and experiment, and revisited in review articles (see citations and a brief explanation of their effect below).

TERMINOLOGY PROBLEM: GIVING A NEW NAME TO A WELL-KNOWN EFFECT
The basic problem is that the terminology "Inverse Current in Coupled Transport (ICC)" was recently invented in the domain of study of "Hamiltonian systems", and it is NOT the terminology usually used in the domain of the nanoscale thermoelectric systems (studied in this manuscript). The terminology for nanoscale thermoelectric systems is that used in bulk thermoelectrics since the 1940s (see Callen's textbook), and has been established for nearly 20 years in nanoscale thermoelectric (see for example the review Benenti and coauthors, Physics Reports, 694, 1 (2017) and many references therein).

WHY THE AUTHORS' DEFINITION OF ICC IS THE SAME AS A NORMAL (NEGATIVE) THERMOELECTRIC RESPONSE
Note, I already said the following since my first report on the manuscript (and the other referee for PRB said similar things), so I do not think that the authors have
read our previous reports carefully.

The authors define an ICC as follows (1st paragraph on page 1) "This could lead to the possibility of Inverse Current in Coupled transport (ICC) when both forces are mutually parallel (i.e. F_E > 0 and F_N > 0), yet one of the induced currents flows against both forces (either J_E < 0 or J_N < 0)."
However, the authors seem to have MISSED my following point in previous referee reports. In a thermoelectric subject to a SMALL positive temperature difference and LARGE positive voltage difference (F_E > 0 and F_N > 0), there is heat flow induced by the Peltier effect. The sign of this heat flow (J_E) is opposite for ANY two systems with opposite thermoelectric response (such as n-type and p-type semiconductors). So at least one of them must have NEGATIVE J_E. The authors claim that this is new, and want to call it an "ICC". Yet, anyone who has read a textbook on thermoelectrics will say that it is well known, and call it "a thermoelectric with negative response".

A standard Peltier refrigerator REQUIRES requires this, and so it is typically made of pairs of thermoelectric's with opposite thermoelectric responses, such as n-type and p-type semiconductors (see wikipedia diagram https://en.wikipedia.org/wiki/Thermoelectric_effect#/media/File:Thermoelectric_Cooler_Diagram.svg, explained in section "Peltier cooling" of https://en.wikipedia.org/wiki/Thermoelectric_effect).
This means that "ICC" is just the AUTHOR'S NEW NAME for a TEXTBOOK thermodynamic response.

The strange thing is that the authors should be well-aware of this, given that the recently published their PRB article on their set-up, and correctly calling it a thermocouple. So they must be aware that thermocouples work on exactly this principle.

These thermoelectric responses have been well-studied in bulk systems since the 1940s, well-studied in nanoscale two-terminal devices since Humphrey & Linke, Phys. Rev. Lett. 94 (2005) 096601], and well-studied in the three-terminal double-dot system since Sanchez and Buttiker Phys. Rev. B 83, 085428 (2011). They were by a number of publications on this three-terminal double-dot system, including an EXPERIMENTS in Nature Nanotechnology 10, 854–858 (2015). Much of the physics was then reviewed in
Benenti and coauthors, Physics Reports, 694, 1 (2017) (see also many references therein). That review clearly explains when a flow will have a negative sign, even though all forces are positive. In short, it occurs
when particles flowing out of a reservoir have energy BELOW the electro-chemical potential of that reservoir. Then we have a heat current in the OPPOSITE direction to the particle current. As these two currents are OPPOSITE, then one of them must be what the authors call a "ICC". Yet, all-previous works on such systems just call this a "thermoelectric response" (which can have either sign, as know since
the 1940s).

The authors' particular set-up (three-terminal double-dot), was already analyzed by Sanchez and Buttiker, and reviewed by Benenti et al's, see their Eqs. (401) and (402). In the notation used in this
manuscript, one can imagine two situations
(a) If dot L's energy, epsilon_L, is ABOVE \mu_a,
then there is a heat current for terminal "a" in
the SAME direction as the particle current
for terminal "a".
(b) If dot L's energy, epsilon_L, is BELOW
\mu_a-\kappa_c, then there is a heat current
for terminal "a" in the OPPOSITE direction as
the particle current for terminal "a".
Thus one can inverse the heat current just by changing the energy-level in dot "a". It is well known that one of these situations has a positive thermoelectic response, and the other has a negative thermoelectric response. However here the authors claim that the positive thermoelectric response is known, while claiming that the negative thermoelectric response is entirely new, and they call it ICC. This is simply a misunderstanding of the physics and the literature. There is no need to give the new name "ICC" to this OLD (and well-known)
effect of negative thermoelectric response.

Requested changes

If the authors wish to publish this paper on this topic, then they should extensively rewrite this manuscript to ensure:
(i) They are not just renaming old effects with
new names. The authors should correctly using
the standard terminology in the domain, so
referees and readers can easily see whether
their results are novel or not.
(ii) They are not presenting results in their previous
article on the same model from last year
(Phys. Rev. B 109, 125124 (2024)).
In other words, any new manuscript should
state what is novel, and remove repetitions of
material already in their 2024 PRB article.

Recommendation

Ask for major revision

---

## Round 2 · Referee Report · Anonymous (Referee 2) · 2025-5-23

Report

The manuscript "Inverse Current in Coupled Transport: A Quantum Thermodynamic Framework for Energy and Spin-polarized Particle Currents" discusses the existence of so-called "inverse currents" in the thermoelectric response of quantum dot systems, following recent works by Wang et al. [3] and Zhang et al. [4,5]. For this, they propose a three-terminal device consisting on two interacting quantum dots (the mentioned previous works discussed two terminals) for allowing a clearer interpretation, though this chose is not clearly motivated in the text. The definition of inverse currents is a bit confusing. The authors use a thermodynamic approach in terms of the particle and energy currents responding to particle and thermal "forces", whose sign is defined based on their contribution to the entropy production: forces are defined as parallel when they have the same sign, which may be misleading, as I discuss below in more detail. With these definitions, the inverse current is defined as one that flows "against" the two forces. However, the difference between this effect and the well known effect of a thermoelectric current whose sign depends on the porperties of the conductor (which is actually the base of a thermocouple) is not clear. Furthermore, in a three terminal device, the entropy production can be written in terms of currents that are not equal, so the mere definition of the inverse current is confusing. Therefore, I cannot recommend this manuscript for publication before these issues are clarified. Please find below a more detailed discussion of my main concerns.

  • The definition of the inverse current is confusing. Even in a simple two terminal device, the thermoelectric current can be either positive or negative depending on the dominance of electrons or holes to the transport. In a quantum dot, this can even be tuned with a gate voltage by sweeping the position of the discrete state across the chemical potential, as routinely done in many experiments (see e.g., Josefsson et al., Nature Nanotechnology 13, 920 (2018)). However, the whole manuscript seems to be based on the existence of such an effect. In the results section, the energy of one of the quantum dots is fixed to be over the chemical potential, and the change of sign in the current is related to the interaction between the two quantum dots being negative. I wonder if the same effect would appear without requiring complicated interactions by just applying a gate voltage.

  • A spin-spin interaction is introduced additional to the Coulomb interaction between the electrons sitting in two different quantum dots. This is done as a parameter, however, the physical motivation of such interaction or its experimetal relevance is not given. It should also be discuss in which cases it can be larger than the Coulomb interaction such that the overall interaction is negative.

  • The definition of the inverse currents is given in terms of the entropy production in equations (25) and (30). For this, the authors chose the particle currents in terminals b and r, and the energy current in terminal b. This expression appears from equation (24) when one uses the conservation of particles and energy, and replaces J_E^a=-(J_E^b+J_E^r), as well as JN^a=-J_N^b and J_N^r=0. However, one could equivalently chose to replace J_E^b=-(J_E^a+J_E^r), which results in a different definition of the currents and the forces. This is essential, because J_E^a and J_E^b are different precisely due to the presence of the third terminal. therefore, the investigated "inverse" current nor it being "parallel" to the forces is not well defined: a temperature bias between a and b may give a positive heat current in a and negative in b, ore the opposite. Also, using energy instead of heat currents is dangerous, as they are not gauge invariant.

  • Related to the above point, how is the the three terminal configuration helping? Even if the temperature T_r is chosen such that it does not appear in the expression for the entropy production, it does not mean that the heat current J_r is zero.

Minor comments:

  • I find the name "force" not appropriate, as these quantities have not the proper dimensions. Indeed, the particle and thermal "forces" have different dimensions. Would "entropic bias" be more appropriate here, as they are defined by the contributions to the entropy production?

  • In page 2, it is said that particle exchange between the two dots is prevented "due to Coulomb blockade". I don't think this is correct. Coulomb blockade only would avoid transitions between a state with one electron in each dot and two electrons in one of them, but would not avoid an electron in the left dot to tunnel to the right one if the later is empty (or viceversa).

  • I think there is a typo in the argument of f^- in eq. (5).

  • Is it correct that the effective chemical potential defined in eq. (10) is the sum of all chemical potentials in the problem?

  • The sign of X and Y defined in equation (42) is argues to be important. How can they be controlled?

  • Efficiencies are defined in equations (56) and (58), but never computed. These are known expressions, so if they are not used in the text, I would suggest the authors to remove them.

  • Typo in the beginning of the last paragraph of the conclusions: "coupld" instead of "coupled".

Recommendation

Ask for major revision

---

## Editorial Decision

awaiting_resubmission